

# Mechanisms controlling giant sea salt aerosol size distributions along a tropical orographic coastline

Katherine L. Ackerman[1], Alison D. Nugent[1], and Chung Taing[2]

[1]Department of Atmospheric Sciences, University of Hawaiʻi at Mānoa , Honolulu, Hawaiʻi
[2]Department of Chemistry, University of California, Berkeley, Berkeley, California

**Correspondence:** Katherine L. Ackerman (klackerm@hawaii.edu)

**Abstract.** Sea salt aerosol (SSA) is a naturally occurring phenomenon that arises from the breaking of waves and consequent bubble bursting on the ocean's surface. The resulting particles exhibit a bimodal distribution, spanning orders of magnitude in size which introduce significant uncertainties when estimating the total annual mass of SSA on a global scale. Although estimates of mass and volume are significantly influenced by the presence of giant particles (dry radius > 1 $\mu$m), effectively ob-

serving and quantifying these particles proves to be challenging. Additionally, uncertainties persist regarding the contribution of SSA production along coastlines, but preliminary studies suggest that coastal interactions may increase SSA concentrations by orders of magnitude. Moreover, our knowledge regarding the vertical distribution of SSA in the marine boundary layer remains limited, resulting in significant gaps in understanding vertical mixing of giant aerosol particles and the specific environmental conditions facilitate their dispersion. By addressing these uncertainties, particularly in regions where SSA constitutes a sub-

stantial percentage of total aerosol loading, we can enhance our comprehension of the complex relationships between the air, sea, aerosols, and clouds.

A case study conducted on the Hawaiian Island of Oʻahu offers insight into the influence of coastlines and orography on the production and vertical distribution of giant SSA size distributions. Along the coastline, the frequency of breaking waves is accelerated, serving as an additional source of SSA production. Furthermore, the steep island orography generates strong and

consistent uplift during onshore trade wind conditions, facilitating vertical mixing of SSA particles along windward coastlines. To investigate this phenomenon, in-situ measurements of SSA size distributions for particles with dry radii ($r_d$) $\geq$ 2.8 $\mu$m were conducted for various altitudes, ranging from approximately 80-650 m altitude along the windward coastline and 80-250 m altitude aboard a ship offshore. Comparing size distributions on and offshore confirmed significantly higher concentrations along the coastline, with 2.7-5.4 times greater concentrations than background open-ocean concentrations for supermicron

particles. These size distributions were then analyzed in relation to critical environmental variables influencing SSA production and atmospheric dynamics. It was found that significant wave height exhibited the strongest correlation with changes in SSA size distributions. Additionally, simulated SSP trajectories provided valuable insight into how production distance from the coastline impacts the horizontal and vertical advection of SSA particles of different sizes under varying trade wind speeds. Notably, smaller particles demonstrated reduced dependence on local wind speeds and production distance from the coastline,

experiencing minimal dry deposition and high average maximum altitudes relative to larger particles. This research not only





highlights the role of coastlines in enhancing the presence and vertical mixing potential of giant SSA but also emphasizes how it is important to consider the influence of local factors on aerosol observations at different altitudes.

# 1 Introduction

The species, sizes, and concentrations of aerosols play critical roles in cloud formation and precipitation processes globally. Out of the several natural aerosol species, sea salt aerosol (SSA) is the most dominant globally by mass (Lewis and Schwartz 2004) and boasts the broadest particle size range, spanning Aitken to ultragiant in dry radius particle size ($r_d$). Studies estimate that sea salt particles (SSPs) account for up to 10% of Aitken mode aerosol particles, 30% of accumulation mode particles, and a majority of all coarse mode particles in a typical marine environment (Zheng et al. 2018). These giant sea salt particles (GSSPs, $r_d > 0.5$ $\mu$m) are formed orders of magnitude less often than their smaller counterparts, however. GSSPs are formed by two primary mechanisms; as jet droplets (0.5 $\mu$m < $r_d$ < 12.5 $\mu$m) produced by the collapse of a bursting bubble on the ocean surface, and as spume droplets (12.5 $\mu$m < $r_d$) through the direct shearing of water from wave surfaces under strong wind conditions (Lewis and Schwartz 2004, p.37). It's noteworthy that while giant and ultragiant aerosol particles ($r_d > 10$ $\mu$m) are the smallest in number concentration, their global concentration fluxes depend heavily on SSA production (Fitzgerald 1991; Zheng et al. 2018), and thus, the interactions between marine clouds and coarse mode aerosols can be attributed to the existence of GSSPs (Jensen and Lee 2008).

Decades of research have been dedicated to quantifying the production of SSA. From one of the first in-situ SSA measurements by Woodcock (1953) to more recent in-situ (Flores et al. 2020) and remote-sensing (Bian et al. 2019) observations of SSA globally, countless efforts have been made to observe, quantify, and characterize SSA production over the open-ocean. SSA is primarily formed through whitecap generation (Monahan et al. 1986), where breaking waves entrain air beneath the ocean surface and generate bursting bubbles that release hundreds of sea water droplets into the atmosphere. To characterize these whitecap interactions, in-situ, remote sensing, and laboratory experiments analyzed how environmental parameters, such as 10 meter wind speeds ($U_{10}$) (De Leeuw 1986; Monahan et al. 1986; Andreas 1998; Gong 2003; Lewis and Schwartz 2004; Clarke et al. 2006; Petelski and Piskozub 2006; Andreas et al. 2008; Norris et al. 2008), sea surface temperature (SST) (Mårtensson et al. 2003; Jaeglé et al. 2011; Zinke et al. 2022), and sea surface salinity (Sofiev et al. 2011; Zinke et al. 2022), contribute to changes in concentrations, masses, and observable SSP size ranges. Despite the large variety of open-ocean SSA production equations (Grythe et al. 2014), a large majority utilize $U_{10}$ wind speeds to represent the primary production mechanism of SSA.

Quantifying the production of SSA in coastal environments faces additional challenges compared to the open-ocean, however, due to the heterogeneity of these regions. Strong regional variations in coastal bathymetry, dissolved surfactants, and organic materials may play a more significant role in wave breaking characteristics and bubble generation compared to the open-ocean. Monahan (1995) hypothesized that the increase in coastal wave breaking would result in a larger whitecap coverage ratio, and therefore SSA production could also be represented by whitecap generation. As a result, $U_{10}$-dependent open-ocean equations were modified to represent the increased whitecap coverage in coastal environments (De Leeuw et al. 2000;



Piazzola et al. 2002; Clarke et al. 2006; Piazzola et al. 2009), and other groups continued to utilize the relationship between
$U_{10}$ and coastal SSA observations more directly (Andreas 1998, 2002; Lewis and Schwartz 2004; O'Dowd and De Leeuw
2007).

The coast experiences a decoupling between wave breaking and local wind speeds, though, as waves may break independently of the local wind speeds due to changes in coastal bathymetry. Therefore, coastal production has a wave-dependent and wind-dependent component that are less related to each other than over the open-ocean. For these reasons, other groups took a
hydrodynamical approach to coastal SSA production and models were developed with closer relationships to wave energetics like wave height (Chomka and Petelski 1997), wave energy dissipation (Van Eijk et al. 2011), and wave slope variance (Bruch et al. 2021). Furthermore, there exists a wind speed threshold for spume droplet production of $> 9 \text{ m s}^{-1}$ (De Leeuw et al. 2000), where winds above this threshold add to the SSA concentrations and winds below it act to the dilute the SSA concentration (Clarke et al. 2006; Hwang et al. 2016). This is because wind speeds that are not actively contributing to production
of SSA only work to increase the volume of air that the SSA are dispersed throughout. The abundant factors to coastal SSA production combined with the lack of observations have resulted in inconsistencies in the primary variable to production in these regions, but currently there are not enough observations to validate any one particular model.

Dynamical features within a sampling environment are also important considerations when observing coastal SSA. Gathman and Smith (1997), Hooper and Martin (1999), and Porter et al. (2003) remotely observed SSA plumes up to 20-25 m
using lidar, while De Leeuw et al. (2000) hypothesized that supermicron particle plume heights could reach upwards of 60 m under moderate $U_{10}$ wind speeds ($6 \text{ m s}^{-1}$). Despite these findings, almost all historical in-situ coastal aerosol studies occurred within the bottom 30 m of the atmosphere, with a majority conducted within the bottom 10 m. When considering the role coastal dynamics may play in the advection of SSPs from their production location, sampling too low to the surface may be an incomplete sample set. There can also be abrupt changes to aerosol sources and sinks, temperature, relative humidity, and
pressure, as well as changes to dynamical properties such as turbulent mixing in the environment as an air parcel transitions from the ocean to coastline (Vignati et al. 2001). The heating differential between the land and ocean can increase vertical velocities over land, generating thermals that rapidly advect SSPs away from the surface and therefore stationary in-situ sampling aparati. Changes in local wind conditions can also affect dynamics in the coastline. Porter et al. (2003) observed the effects of very weak trade wind speeds ($1\text{-}2 \text{ m s}^{-1}$) on the generation of near-shore thermals, causing offshore SSA plumes to be rapidly
vertically mixed to hundreds of meters high before reaching the coastline. Orography can also play a role as steep terrain can induce strong vertical velocities and reduce horizontal velocities of onshore winds as a result of upwind mountain blocking (Minder et al. 2013). Therefore, impacts from the sampling environment are important to consider, as a significant portion of vertically mixed SSA particles may be missed if coastal samples are collected within a small altitude range.

Overall, the limited number of observations of coastal SSA contribute to large uncertainty in the processes that control
production in this region. Previous coastal studies utilized mass, total number concentration, and range-restricted SSA-size distributions (SSA-SD) within the surface layer of the atmosphere to characterize how environmental variables contribute to coastal SSA concentrations. In this study, we aim to build upon previous coastal SSA research with our novel sampling method





to observe the impacts of the environment on GSSP concentrations. We utilize in-situ measurements of SSA-SDs from 80 m - 650 m along a tropical orographic coastline to explore three questions:

1) How do SSA-SDs change from the open-ocean to the coastal environment?,

2) What environmental variable is most significantly correlated to changes in SSA-SDs? Wind speeds, or sea state?, and

3) How do dynamics within our sampling location contribute to variation the SSA-SDs across different environmental conditions and altitudes?

The focus of this paper remains on expanding our understanding of current coastal SSA production, as well as understanding
mechanisms that control their transport in a coastal orographic region. Our ability to sample the largest end of the SSA-SDs provides important insight into these statistically understudied GSSPs.

## 2   Methods

### 2.1   Coastal SSA Samples

#### 2.1.1   The Mini-GNI Instrument

The uniqueness of this aerosol collection method lies in the versatility of the mini-Giant Nucleus Impactor (mini-GNI) (Fig. 1a). The mini-GNI is an accessible and affordable sampling apparatus that collects GSSPs; full details on the aerosol collection methods are provided in Taing et al. (2021). The mini-GNI contains a singular polycarbonate slide that collects aerosol particles when exposed to a free air stream. A door covers the slide to protect it from collecting particles during transit, allowing sample collection at discrete altitudes, time intervals, and geographic locations. In-situ measurements of pressure (P), relative humidity
(RH), temperature (T), as well as the status of the door (open or closed) are recorded at 1 Hz throughout sampling, enabling accurate measurement of the environmental conditions where aerosol particles are collected. Lastly, all of this information results in a sea salt aerosol size distribution (SSA-SD) for each slide, where the observable particle range is determined by the collision efficiency of each particle.





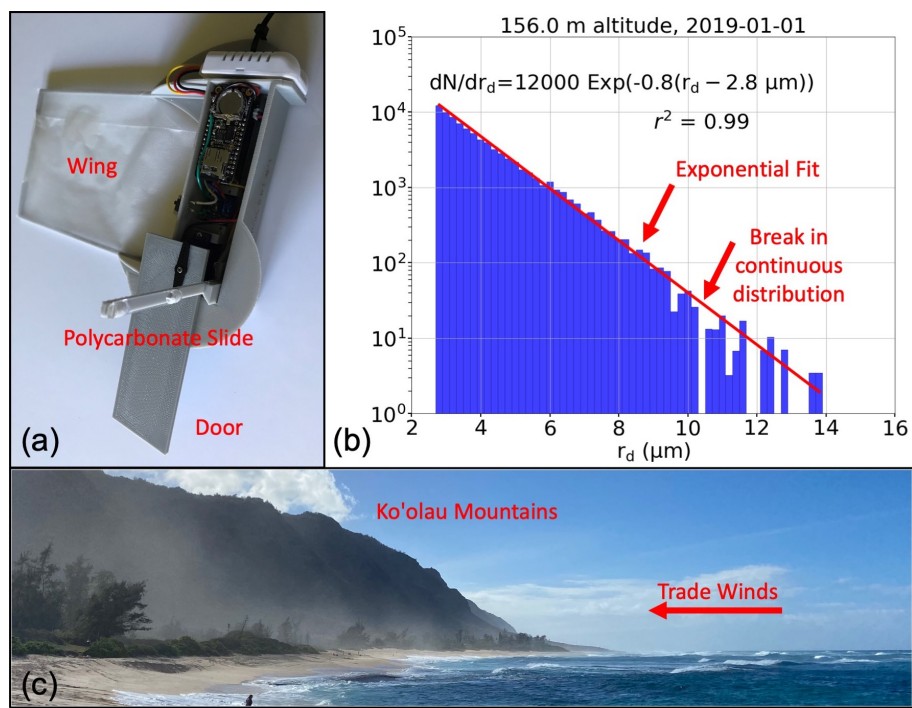

**Figure 1.** (a) A mini-GNI instrument is shown in its open position with the clear polycarbonate slide exposed. The door swivels shut to protect the slide from collecting SSPs during transit, while the wing covered with a thin plastic sheet helps keep the slide perpendicular to the wind during flight. The T and P sensors are within the body of the instrument, while the RH sensor is pictured outside the body on the top of the instrument. (b) An example SSA-SD from January 1st, 2019 at 156 m altitude is shown. The red line shows the exponential fit, fitted with Eq. 1, and an $r^2$ value to show how well this fit represents the observed size distribution. The exponential fit is fitted up to the break in continuous SSA-SD (e.g. up to 10.2 $\mu$m). (c) A photo of the Ko'olau Mountains with coastal SSP generation in the foreground. The trade winds are blowing onshore, bringing open-ocean and coastally generated SSPs towards the mountains, which are rapidly lifting them.



Calculations of collision efficiency (CE) are dictated by two factors: the relative airspeed the slide experiences during sam-
pling and the size of the deliquesed SSP. For all land-based samples within this study, the mini-GNI is deployed on a stationary
system, meaning the relative wind speeds are the true wind speeds at that altitude and geographic location. When the mini-GNI
is deployed on a non-stationary system (i.e. a moving boat), the relative wind is based on the true wind and the moving sys-
tem's motion. The mini-GNI can orient itself parallel to the wind stream, thereby keeping the slide perpendicular to the wind
direction and maximizing the CE. The weakest wind observation dictates what particle sizes can be compared across samples,
and only particles with a collision efficiency of $> 40\%$ are used for analyses. The weakest observed wind speed at altitude for
samples in this study is $5.6$ m s$^{-1}$ meaning the smallest observable particle has an r$_d = 2.8$ $\mu$m. Full details on CE calculations
can be found in Taing et al. (2021).

Due to the high hygroscopic nature of SSA, the RH of the environment also plays a significant role in the CE of particles.
Higher RH results in greater condensational growth of SSPs, giving them larger radii, mass, and inertia than SSPs of the same
dry radius at lower RH. In-situ measurements of RH, T, and P from the mini-GNI are used to convert a deliquesced SSP to r$_d$,
while calculated wind speeds aloft (see Sect. 2.3.2) are utilized for the CE. Lewis and Schwartz (2004, p. 53) consider RH to
be the most significant variable to SSP growth, and this study assumes the growth of SSPs is analogous to the growth of pure
NaCl particles (Lewis and Schwartz 2004, p. 54) and that these SSPs have reached their equilibrium radius (Taing et al. 2021).

After collection, the aerosol slides are analyzed using the microscope methods detailed in Jensen et al. (2020). The resulting
SSA-SDs are fit with an exponentially decreasing function using the least squares fit method of $\ln(dN/dr_d)$ to r$_d$ (Taing et al.
2021). Here, a modified form of this equation is used:

$$\frac{dN}{dr_d}(r_d) = r_{2.8}e^{-B(r_d - 2.8\mu m)} \tag{1}$$

where $dN(r_d)/dr_d$ is the concentration (N) m$^{-3}$ $\mu$m$^{-1}$ at the bin (width $0.2$ $\mu$m) centered on the dry particle radius (r$_d$). r$_{2.8}$
is the number concentration m$^{-3}$ $\mu$m$^{-1}$ for a dry particle with a radius of $2.8$ $\mu$m, the smallest observable SSP size in this
study. r$_d$ is therefore the dry particle radius valid for SSPs $\geq 2.8$ $\mu$m, and B is the inverse of the characteristic radius for the
size distribution with B $> 0$. Inclusion of the $-2.8$ $\mu$m in the exponent is implicit in the remainder of the equations used in this
study due to the sampling range limitations. This function was fit to each size distribution from $2.8$ $\mu$m through the largest
sized particle for bins that were continuous; i.e., the function stops fitting to particles when there was a break in observed SSP
bins (Fig. 1b, red line). These exponential fits are then used in this study to represent the SSA-SDs for given samples if the r$^2$
values of the fits were $> 0.90$.

### 2.1.2   COAST Samples

A ground-based kite platform (Taing et al. (2021), Fig. 1a) sampled SSA-SDs along the Eastern Coast of Oʻahu at Kaupō Bay
(21°18' 54" °N, 157°39' 43.2" °W, Fig. 2). The kite platform was set-up approximately 33 m inland from the shore break and
4 m above the mean sea surface elevation. At this location the trade winds blow onshore, lofting the kite along the southern
portion of the Koʻolau mountain range (Fig. 2). Every deployment utilized three to five mini-GNIs to simultaneously sample at
different altitudes, as well as two iMet XQ2 instruments; one just below the kite and the other on the ground surface to measure



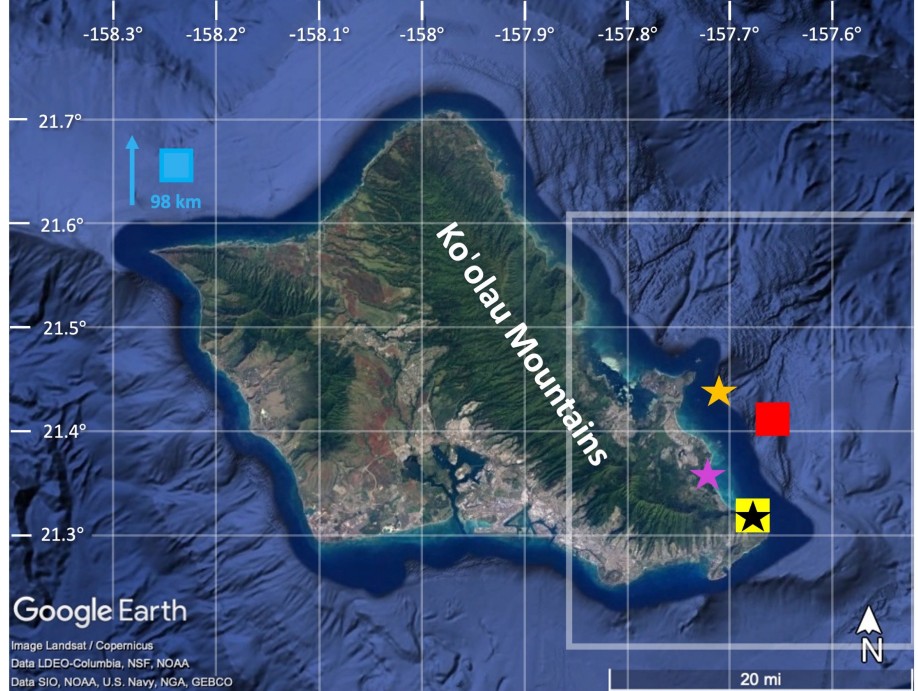

**Figure 2.** Sampling and data locations are marked on a map of Oʻahu. The samples from the open-ocean cruise (OCEAN) were taken 98 km north of the western tip of Oʻahu marked by the blue square, while the coastal samples (COAST) were taken within the yellow square at Kaupō Bay. W53 samples are estimated to come from the region around the red square based on diagrams in Woodcock (1953). The wave state information came from Mōkapu buoy (orange star), while wind data came from Bellows (purple star) and WeatherFlow's Makapuʻu station (black star). The WRF domain is approximated by the white box but extends further to the south and east than this image. Base image is from NOAA, used under licence from Google Earth.

P, T, and RH (InterMet 2017). A Kestrel 5500 weather station determined preliminary wind speeds for the day, and recorded approximate 2 m wind speeds throughout the duration of sampling (Kestrel 2015). The individual SSA samples (n = 77) were observed on 11 different days over an approximately 1 year period(Dec 2018 - Sep. 2019) and occurred on days with moderate
(3-8 m s$^{-1}$) onshore trade winds (Table 1). Additional details on the collection methods and processing can be found in Taing et al. (2021).





**Table 1.** A total of 77 coastal samples from 11 sampling days, as well as 8 open-ocean samples from 3 sampling days (HOT), were utilized in this study. Altitudes range from 86 to 638 m. Sampling date (YYYY-MM-DD), the sample #, number of samples taken on each sampling date (n), sampling altitude minimum and maximum (m), and averaged values of surface P (hPa), RH (%), T from samples (°C), $U_{10}$ (m s$^{-1}$), $U_{10}$ wind direction (°), significant wave height ($H_s$, m), mean wave period (Per., s) and SST (°C) are shown in the table. For atmospheric state variables, the averages are taken across all sampling altitudes recorded by each mini-GNI for the duration of sampling except for surface pressure, while ocean variables are averaged across the sampling duration from their collection locations. Wind direction was not available for the three open-ocean samples.

| Date | Sample # | n | Alt. Min. | Alt. Max. | P | RH | T | $U_{10}$ | $U_{10}$ Dir. | $H_s$ | Per. | SST |
|---|---|---|---|---|---|---|---|---|---|---|---|---|
| 2019-01-14 | HOT 1 | 2 | 135 | 143 | 999 | 82 | 22.3 | 4.66 | n/a | 1.24 | 6.78 | 24.43 |
| 2019-01-16 | HOT 2 | 1 | 166 | 166 | 992 | 82 | 22.1 | 5.19 | n/a | 2.37 | 6.50 | 24.85 |
| 2019-01-17 | HOT 3 | 5 | 82 | 236 | 994 | 74 | 21.3 | 5.78 | n/a | 1.76 | 7.69 | 24.66 |
| 2018-12-05 | 1 | 5 | 86 | 253 | 1016 | 76 | 24.0 | 6.10 | 70 | 2.08 | 7.88 | 25.62 |
| 2019-01-01 | 2 | 4 | 88 | 241 | 1016 | 82 | 23.9 | 7.08 | 80 | 2.32 | 6.47 | 24.78 |
| 2019-04-13 | 3 | 7 | 140 | 456 | 1022 | 75 | 23.6 | 7.15 | 54 | 2.89 | 7.06 | 24.00 |
| 2019-04-23 | 4 | 6 | 136 | 448 | 1021 | 73 | 23.8 | 5.83 | 62 | 1.72 | 5.92 | 24.31 |
| 2019-06-16 | 6 | 2 | 198 | 362 | 1017 | 68 | 26.4 | 5.62 | 68 | 1.55 | 5.10 | 25.75 |
| 2019-07-31 | 7 | 9 | 158 | 471 | 1016 | 85 | 24.7 | 6.92 | 63 | 2.26 | 6.29 | 26.75 |
| 2019-08-15 | 8 | 12 | 90 | 364 | 1017 | 72 | 27.6 | 5.78 | 69 | 1.78 | 5.48 | 27.33 |
| 2019-08-20 | 9 | 11 | 173 | 498 | 1017 | 75 | 27.2 | 5.26 | 64 | 1.61 | 5.29 | 27.66 |
| 2019-08-22 | 10 | 12 | 151 | 638 | 1018 | 76 | 26.9 | 5.27 | 50 | 1.33 | 5.21 | 27.60 |
| 2019-09-10 | 11 | 9 | 153 | 635 | 1019 | 79 | 25.8 | 6.16 | 51 | 1.54 | 5.80 | 27.72 |





## 2.2 Open-Ocean SSA Samples

### 2.2.1 OCEAN Samples

Using the same kite methodology as the coastal samples, the kite platform was deployed aboard the Hawaiian Oceanographic
Time-series (HOT) cruise #309 to collect open-ocean SSA samples. The HOT cruise utilizes the Kilo Moana research vessel
and travels to Station ALOHA, approximately 98 km due north of the western-most point of Oʻahu (Fig. 2) in approximately 4
km deep water. During transects to, from, and around Station ALOHA, a total of nine samples were collected from 80-240 m
altitudes across three separate days (Table 1).

### 2.2.2 Historical Samples

Historical open-ocean SSA-SDs collected over the ocean from Woodcock (1953) were reconstructed for comparison to samples
in this study. Woodcock (1953) used a similar impaction method to the mini-GNI, exposing several slides from an aircraft at
a variety of altitudes northeast of Kaupō Bay, Oʻahu (Fig. 2). The SSA-SDs were plotted as inverse cumulative concentrations
($N_c$) along with records of the sampling day's wind force - a qualitative measurement of wind speed and sea state based on the
Beaufort Wind Scale. The Wind Force values were converted to approximate $U_{10}$ wind speed ranges (Britannica 2023), and
the $N_c$s from Woodcock (1953)'s Figure 1 were converted from formation radius size at 99% RH to $r_d$ for comparison to this
study. All further references to samples from Woodcock (1953) will be referred to as W53.

## 2.3 Environment Data

### 2.3.1 Sea State

Sea state data like significant wave height ($H_s$) and SST came from the Pacific Islands Ocean Observing Systems (PacIOOS)
Mōkapu buoy (Coastal Data Information Program (CDIP) et al. 2000), located 4.5 km offshore of the windward coastline at
a water depth of 86 m (Fig. 2). Depending on the swell conditions, this buoy represents intermediate to deep water waves,
with the average wave height to wavelength ratio maintaining deep water wave classification. This buoy remains the closest
geographically to Kaupō Bay, and represents approximate open-ocean sea state conditions for the eastern portion of Oʻahu.

### 2.3.2 $U_{10}$ Calculations

In February of 2020 a private 10 m anemometer was installed at Kaupō Bay by WeatherFlow (R.M. Young Company 2016),
approximately 5 months after the completion of SSA sampling. A linear regression on almost two years worth of data was
conducted between Bellows' 12 m wind speeds and WeatherFlow's 10 m wind speeds for trade wind days (wind directions
between 30-90°), resulting in an overall Pearson's r = 0.904. For comparison, the in-situ 2 m Kestrel samples only correlate to
Bellows 12 m wind speeds with a Pearson's r = 0.681. Therefore, a relationship between the WeatherFlow 10 m wind data and
the Bellows 12 m wind data was derived to approximate historical wind speeds for the SSA sampling dates.





Finally, following methodology from Taing et al. (2021), these wind speeds (u(z)) at the sampling altitudes aloft ($z$) were calculated using these WeatherFlow $U_{10}$ wind speeds ($u_r(z_r)$) applied to a logarithmic dependence of wind speed with altitude from Arya (2001):

$$\frac{u(z)}{u_r(z_r)} = \frac{\ln(z/z_0)}{\ln(z_r/z_0)} \tag{2}$$

The assumed surface roughness length ($z_0$) was 0.001, consistent with coastal location measurements (Arya 2001). These
wind speeds were then used to calculate the CE for all SSA particle sizes as well as the total air volume sampled. For the purpose of this paper, any reference to $U_{10}$ wind speeds for coastal SSA samples will be to these calculated wind speeds for the Kaupō Bay area.

### 2.3.3 Vertical Environment Profiles

For every kite deployment, an iMet XQ2 instrument was attached approximately 25-m below the kite. The iMet XQ2 samples
T, P, RH, geographic position (latitude, longitude), and altitude (z) at 1 Hz frequency, for the entire sampling duration of 2-3 hours. Vertical profiles of the environment were collected during the kite ascent and descent, which occurred approximately three times per sample date. T, RH, and P were averaged 1 m altitude increments for the sampling period and used to calculate the terminal fall velocities of six different SSPs as they change with altitude and changing environmental conditions.

### 2.4 SSA Trajectory Modeling

### 2.4.1 WRF Model Set-Up

The Weather and Research Forecasting (WRF) model V4.0 provided wind field estimates across high spatial resolution for two sampling days in this study (Skamarock et al. 2019). Only u, v, and w-speeds were utilized from these simulations to supplement wind information for a high resolution 3-dimensional fluid flow over complex terrain. A single square domain was chosen from 21°36' 49" N, 157°52' 5.52" W to 21°11' 9.92" N 157°24' 47.16" W resulting in 250 by 250 grid points
with 200 m horizontal resolution (Fig. 3a). The domain was forced with National Centers for Environmental Prediction's (NCEP) 1° Model Global Tropospheric Analysis with custom 200-meter topography to realistically represent updraft velocities created by the steep orographic gradients of the Koʻolau Mountains. Additionally, custom $\eta$-values were chosen for the lower portion of the atmosphere, with the lowest 1.5 km represented by approximately 40 m vertical levels, followed by 100 m vertical levels through 5 km. The Thompson microphysics scheme was used (Thompson and Eidhammer 2014), along with
RRTMG for longwave and shortwave radiation, and the Shin and Hong (2015) PBL scheme for the grid resolution. The diffusion option was evaluated in physical space (stress form), and the horizontal Smagorinsky first order closure was used for computing the K coefficient. Because the terrain slopes are > 45° in some regions, the EPSSM parameter was tuned to dampen vertically propagating sound waves and prevent model errors. Other namelist parameters were in agreement with NCAR recommendations for WRF runs with gridspaces between 100 m and 1 km.



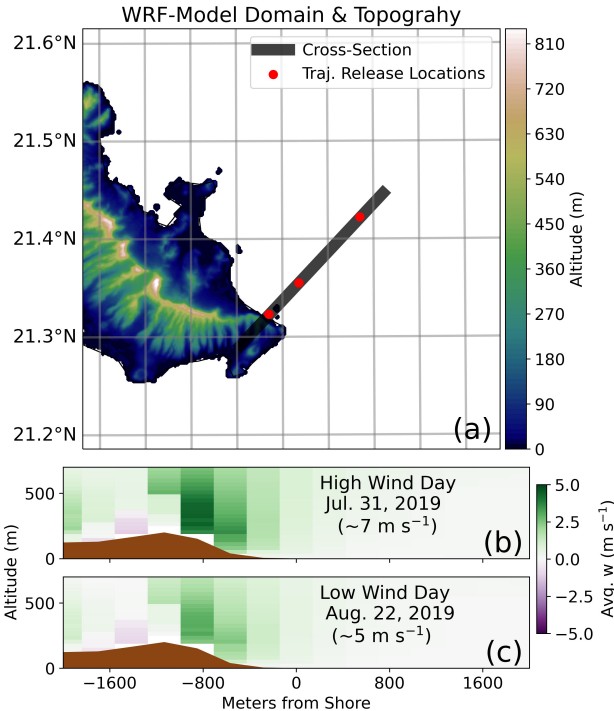

**Figure 3.** (a) The complete WRF domain is shown with the topography of the Koʻolau Mountains. The black line represents the 3-hour averaged vertical cross section taken for plots (b) and (c), and the red dots represent the production locations for the trajectories in Sect. 6. The average simulated vertical velocities (w) within the cross section are shown on a (b) high wind day (average u-v component of 7 m s$^{-1}$ observed in-situ) and a (c) low wind day (average u-v component of 4 m s$^{-1}$ observed in-situ) for comparison.

Two simulations were completed based on wind data from our sampling dates; one relatively high wind speed day (HW, July 31st 2019), with a 7 m s$^{-1}$ average in-situ $U_{10}$, and one relatively low wind speed day (LW, August 22nd 2019), with a 5 m s$^{-1}$ average in-situ $U_{10}$. After approximately 24 hours of model spin-up time, a vertical transect was taken perpendicular to the coastline (Fig. 3a), where the u, v, and w fields were averaged for a 3 hour period for our in-situ sampling period (Fig. 3b and c). For the total space of the transects, the high (low) wind simulation had an average updraft speed of 0.40 m s$^{-1}$

(0.22 m s$^{-1}$), an average u component of -8.54 m s$^{-1}$ (-5.34 m s$^{-1}$) and average v component of -5.68 m s$^{-1}$ (-3.84 m s$^{-1}$). Differences in updraft speeds between these two days along the coastline varied from as little as 0.1 m s$^{-1}$ to over 1.5 m s$^{-1}$.





### 2.4.2 SSA Fall Velocity Calculations

Fall velocities that include hygroscopic particle growth with altitude were calculated for various sized GSSPs. The fall velocities ($v_\infty$) utilized equations from Table 1 in Beard (1976) with the addition of dry salt mass for each GSSP, as the original equations treated the falling droplets as pure water. The fall velocities for six GSSPs ($r_d$ = 2.8 $\mu$m, 5.0 $\mu$m, 7.4 $\mu$m, 9.8 $\mu$m, 12.2 $\mu$m, and 15.6 $\mu$m) were calculated from the surface through cloud base at 1 m increments using the averaged vertical profiles from the iMet XQ2 for the high and low wind day. Reference to these six particle sizes will be in the form $r_d$ followed by a subscript of their dry particle size in microns, for example $r_{d2.8}$, and are not to be confused with the concentration variable $r_{2.8}$ defined in Sect. 2.1.1. Figure 4 shows how particle growth and fall velocities change with altitude for the low wind day (August 22nd, 2019). Altitude is effectively an analog for RH in this case, as the observed RH profiles with altitude were approximately linear for both days. The condensational growth equation from Lewis (2008) shows hygroscopic growth of a SSP given its dry radius and RH:

$$r_{RH} = 1.08 r_d \left(1.1 + \left(\frac{1}{1 - RH + \left(\frac{1.1E-9}{1.08 r_d}\right)^{\frac{3}{2}}}\right)^{\frac{1}{3}}\right) \tag{3}$$

where $r_{RH}$ is the deliquesced particle size at that relative humidity. Because the RH remains $\geq 70\%$, we assume that the GSSPs are spherical to remove complications created by amorphous droplet shapes.

### 2.4.3 SSA Trajectory Calculations

Lastly, trajectories for GSSPs sized $r_{d2.8}$, $r_{d7.4}$, and $r_{d12.2}$ were simulated using the WRF wind profiles and calculated fall velocities from both wind days. Three production locations (represented by the red dots in Fig. 3; 100 m (Coastline), 2600 m (Intermediate Zone), and 5800 m (Open-Ocean) offshore) were chosen to study how distance from the coastline affects each particle size's horizontal and vertical mixing potential the high and low wind day. 100 trajectories were run for every particle size, production location, and wind speed day, resulting in a trajectory range for each scenario (Fig. 11). GSSPs were released between 10 m and 25 m, decreasing exponentially with altitude from 10 m in agreement with observations from Gathman and Smith (1997), Hooper and Martin (1999), and Porter et al. (2003). Because the GSSPs were released at 10 m and above, we assume they are already in equilibrium with the ambient RH (Lewis and Schwartz 2004).

As the GSSPs moved throughout the domain, the trajectory was tracked until the particle reached sufficiently past our kite sample location or when the particle reached an altitude of 0 m, for which we assume the GSSP has been removed from the atmosphere via dry deposition. Stochastic variation within two standard deviations of the WRF wind simulations were introduced to each trajectory's u, v, and w field at each time step, with each deviation chosen based on the probabilities of that deviation given a Gaussian distribution centered on the average speed for each wind component. These variations were designed to represent the natural variability in the wind field expected in real life.

Trajectories therefore account for the growth in deliquesed SSP radii based on RH changes with altitude, consequent change in $v_\infty$ from changes in atmospheric state variables with altitude, and the pseudo-natural variation of u, v, and w-speeds at all





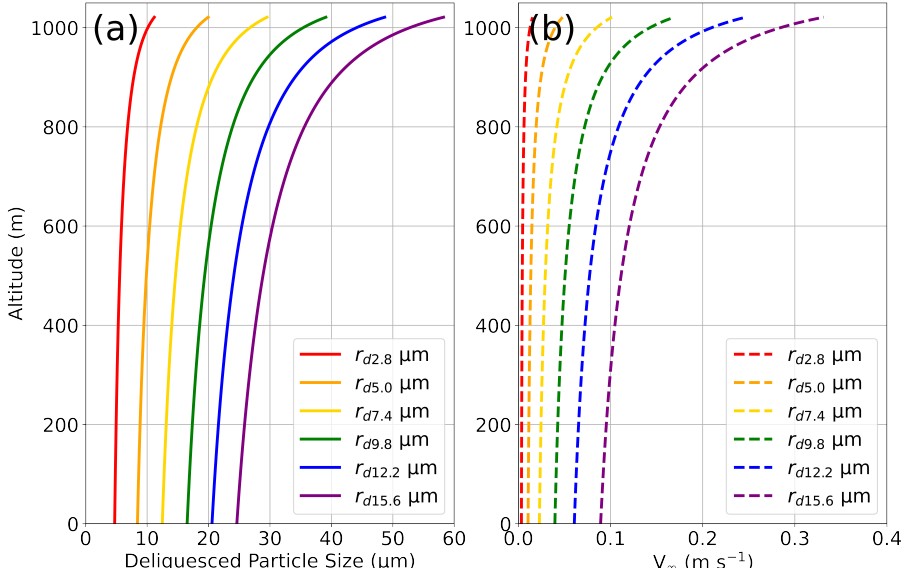

**Figure 4.** The (a) deliquesced particle radius (um) and resulting (b) terminal fall velocity ($v_\infty$, m s$^{-1}$) for six GSSPs are shown, colored by their $r_d$ for the low wind day, August 22, 2019. The hygroscopic growth shown in (a) utilizes the iMet XQ2 RH vertical profiles from the sample date to show how GSSPs grow as they ascend from the surface up to cloud base at 1050 m. SSP radii more than doubles from the average surface RH (70%) to just below cloud base (98%), and the mass of the deliquesced SSPs grow approximately 4.5 times. The calculation of the terminal fall velocity combines the change in mass, radius, and atmospheric state variables with altitude.

particle positions. The average maximum altitude (AMA) was then calculated for all particles for each production location and wind speed day to represent the relative differences in altitude achieved by these trajectories. AMA, however, is artificially
limited by choosing to evaluate an area close to the kite sampling, and therefore does not represent the total potential altitude range these particles may reach beyond the domain in this study.





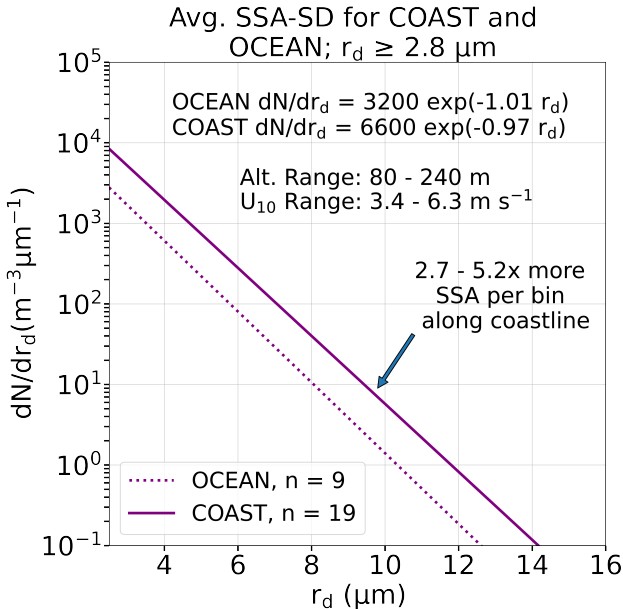

**Figure 5.** Altitude and environment averaged SSA-SDs are plotted for samples taken from the windward coastline (COAST, solid purple line) and the open-ocean (OCEAN, dotted purple line) north of Oʻahu. Both SSA-SDs contain samples between 80-240 m altitude and 3.4-6.3 m s$^{-1}$ wind speeds. The COAST sample has more GSSPs at all observable radii, with a larger multiplying factor as GSSP radii increase. This is also echoed in the COAST size distribution function (dN/dr$_d$) which has a larger r$_{2.8}$ value and less steep slope when compared to the OCEAN size distribution function.

## 3 COAST vs. OCEAN SSA-SDs

### 3.1 Cumulative Concentrations in the Coast vs. Open-Ocean with the Mini-GNI

To assess the impacts of the coastline on SSA-SDs, this study compared coastal SSA-SDs (COAST) and open-ocean SSA-SDs
(OCEAN) within the same $U_{10}$ (3.4 - 6.3 m s$^{-1}$) and altitude (80-250 m) ranges. $U_{10}$ is largely considered the dominant factor to open-ocean SSA production, either indirectly through the generation of whitecaps or directly through spume droplet production at higher wind speeds (De Leeuw et al. 2011). At the smallest observable $r_d$, COAST concentrations m$^{-3}$ $\mu$m$^{-1}$ are approximately 2.7 times higher than OCEAN concentrations m$^{-3}$ $\mu$m$^{-1}$. As $r_d$ increases, average COAST GSSP concentrations gradually increase to 5.2 times greater than average OCEAN concentrations (Fig. 5), resulting in a slight decrease in
B between the OCEAN and COAST samples. This decrease in B also illustrates that the average COAST sample observes a larger total range of GSSP sizes, approximately 2 $\mu$m larger, than the average OCEAN $r_d$ range.



## 3.2 Cumulative Concentrations vs. Historical W53

The OCEAN samples were limited in number and environmental variety, so historical SSA-SDs measured during the 1950s by Woodcock (1953) were used to compare samples over a greater range of observations. For 3.4-5.4 m s$^{-1}$ U$_{10}$ range, the average

OCEAN N$_c$ remains close to W53 N$_c$ (Fig. 6a), demonstrating that OCEAN samples in this study are well matched to historical open-ocean observations. For COAST observations, however, N$_c$s in both U$_{10}$ ranges exceed the W53 N$_c$s. Furthermore, the COAST samples only observed wind speeds up to 7.9 m s$^{-1}$ and still have a larger N$_c$ than W53, which has a total wind speed range up to 10.4 m s$^{-1}$. Additionally, the difference between COAST N$_c$s and W53 N$_c$s remains consistent at all GSSP radii, indicating that COAST samples in this study have a greater concentration of all GSSP sizes.

Next, two W53 N$_c$s at discrete altitudes (561 m and 518 m from Fig. 3 and 4 in Woodcock (1953)) and U$_{10}$ ranges were compared to averaged COAST samples at similar altitudes and U$_{10}$ ranges (Fig. 6b). Only two samples from Woodcock (1953) were useful for this comparison, as they were the only two discrete samples in Hawaiʻi confirmed to be below cloud base. Sampling above or below cloud base is an important distinction due to recent findings that SSA concentrations taken above cloud base do not correlate well to changes in environmental conditions like wind speeds above the ocean's surface (Zheng

et al. 2011). The COAST samples lack observations between 498-614 m, and therefore averaged N$_c$s were calculated for samples taken between 450 m and 638 m. Overall, COAST N$_c$s are greater than W53 N$_c$s for both the 3.4 - 5.4 m s$^{-1}$ and 5.5 - 10.4 m s$^{-1}$ U$_{10}$ ranges. For the stronger U$_{10}$ range, the COAST N$_c$ remains larger than W53 despite lacking observations for U$_{10}$ for the upper range observed in W53 (7.9 - 10.4 m s$^{-1}$). Unlike Figure 6a, these upper altitude COAST samples converge on W53 samples as r$_d$ decreases. We hypothesize that the coastline observes increased vertical mixing of larger GSSPs than

found over the open-ocean at these altitudes, and that concentrations of smaller GSSPs aloft are more similar to open-ocean observations. These differences may be due the negligible updrafts speeds required to mix a 3 $\mu$m GSSP versus those required of an ultragiant SSP (Fig. 4b).





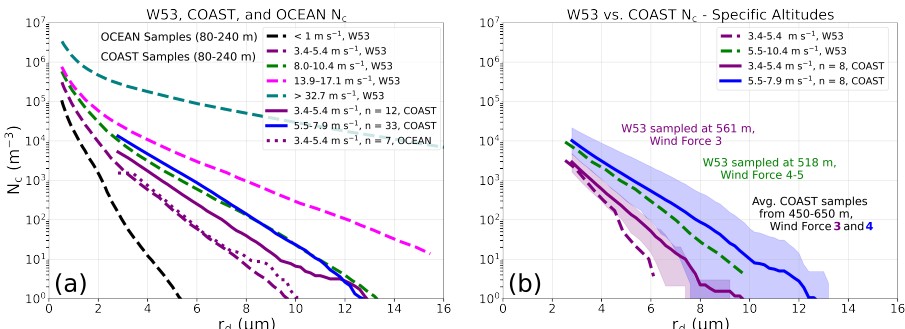

**Figure 6.** Inverse cumulative concentrations ($N_c$) were used to compare how concentrations of SSA change from the largest to the smallest observable SSA bin ($r_d \geq 2.8\ \mu$m). Plot (a) shows the averaged cumulative concentrations measured by W53 (dashed lines) for discrete wind speed ranges, with our COAST (solid lines) and OCEAN (dotted line) samples for the same altitude range (80 - 240 m). Plot (b) shows two specific W53 samples at discrete altitudes (561 m and 518 m from Fig. 3 and 4 in Woodcock (1953)) and wind ranges to averaged COAST samples for the same altitude and similar $U_{10}$ ranges. The shaded regions show the range between the smallest and largest COAST $N_c$ within each $U_{10}$ range.





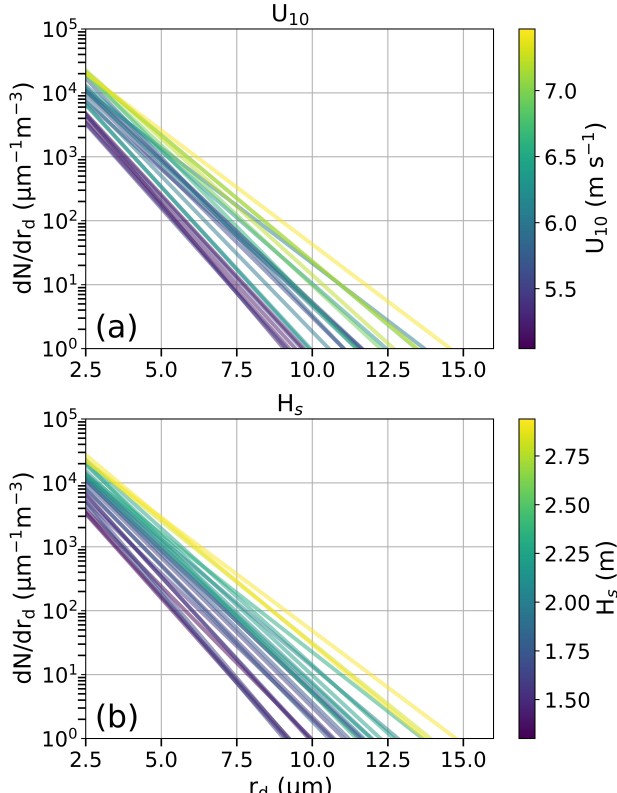

**Figure 7.** SSA-SDs were binned and averaged by two environmental parameters - (a) 10 m wind speed, $U_{10}$, m s$^{-1}$, and (b) significant wave height, $H_s$, m - to visualize the organization of these size distributions to changes in the environment. SSA-SDs show strong positive organization to $U_{10}$ and $H_s$.

## 4 Environmental Dependencies of COAST SSA-SDs

### 4.1 SSA-SDs and Environment Variables

SSA production, and subsequently atmospheric concentrations of SSA, are intrinsically tied to interactions between the air and sea. To understand how the environment impacts the natural variability of SSA, SSA-SDs were plotted for the complete range of $U_{10}$ and $H_s$ observed in this study (Fig. 7). The color of each size distribution represents discrete values for both environmental variables, meaning the more organized the color spectrum, the more correlated SSA-SDs are to changes in that environmental variable. There are strong positive correlations between the organization of SSA-SDs to both $U_{10}$ and $H_s$; as

$U_{10}$ and $H_s$ increase, the overall concentrations and slope of the SSA-SDs become gentler, indicating higher concentrations of GSSPs and a wider range of GSSPs with increases to both environmental variables. SSA-SDs organized by $U_{10}$, however, appear to have slightly higher variability than $H_s$.



**Table 2.** A table containing the $U_{10}$ Pearson's r, $H_s$ Pearson's r, $\rho_{U_{10}}$, and $\rho_{H_s}$ are shown for SSA-SD parameters $r_{2.8}$, B, and $y_{10}$. The $\rho_{U_{10}}$ is the partial correlation of these variables to $U_{10}$ in the absence of $H_s$, while $\rho_{H_s}$ is the partial correlation of these variables to $H_s$ in the absence of $U_{10}$. Bold values indicate strong positive correlations.

| SSA-SD Variable | $U_{10}$ Pearson's r | $H_s$ Pearson's r | $\rho_{U_{10}}$ | $\rho_{H_s}$ |
|---|---|---|---|---|
| $r_{2.8}$ | **0.86** | **0.91** | 0.43 | **0.65** |
| B | **0.68** | **0.79** | 0.03 | **0.55** |
| $y_{10}$ | **0.74** | **0.80** | 0.20 | 0.47 |

## 4.2 SSA-SD shape parameters by $U_{10}$ and $H_s$

To further investigate these relationships' specific impacts on SSA-SD shape parameters, $r_{2.8}$, B, and the cumulative concentra-
tions of ultragiant SSPs ($y_{10}$, $r_d > 10\ \mu$m) were plotted and colored by the average value of $U_{10}$ and $H_s$ for each sample (Fig. 8). Every subfigure in Figure 8 shows a moderate to strong visual correlation between the SSA-SD shape parameters, $U_{10}$, and $H_s$. Given the high correlation between $U_{10}$ and $H_s$ (Pearson's r = 0.85), partial correlations determine the linear dependence between SSA-SD shape parameters and $U_{10}$ or $H_s$ while isolating the influence to only one environment variable. The partial correlation is therefore calculated as:

$$r_{ab.c} = \frac{r_{ab} - r_{ac}r_{bc}}{\sqrt{1 - r_{ac}^2}\sqrt{1 - r_{bc}^2}}. \tag{4}$$

Variable $a$ is any of the three SSA-SD shape parameters, while variables $b$ and $c$ are $U_{10}$ and $H_s$ for correlations isolating the relationship to $U_{10}$, and the inverse when isolating the relationships to $H_s$. Therefore, $r_{bc}$ is always the Pearson's r correlation between $U_{10}$ and $H_s$, which is 0.85 for this study. The other two Pearson's r correlations, $r_{ab}$ and $r_{ac}$, represent the correlation between the two subscripted variables, and $r_{ab.c}$ represents the partial correlation between variables $a$ and $b$ without the
controlling variable $c$.

Figure 8a and b show moderately positive correlations between $r_{2.8}$ and B. Positive correlations are also seen between $r_{2.8}$ and B and environmental variables, and maintain strong Pearson's r correlations (Table 2). When partial correlations ($\rho$) are applied, however, the correlation between $U_{10}$ and $r_{2.8}$ is reduced to a moderate correlation while the partial correlation between $H_s$ and $r_{2.8}$ remains strong. The differences in partial correlations indicates that a majority of the environmental
correlation to $r_{2.8}$ belongs to changes in $H_s$ rather than $U_{10}$, and that the initial strength of the Pearson's r value between $r_{2.8}$ and $U_{10}$ may come from the inherent relationship between $U_{10}$ and $H_s$ in this study. A more drastic pattern emerges with B; the partial correlation between $H_s$ and B remains strong, while the partial correlation between $U_{10}$ and B is entirely removed. B sets the shape of the SSA-SD, meaning a good SSA production equation should be represented by an environmental variable that captures changes in not only concentration, but the concentration chances across different radii bins. Both of these partial
correlations support that $H_s$ is a strong environmental control than $U_{10}$ on coastal SSA-SD shape parameters.




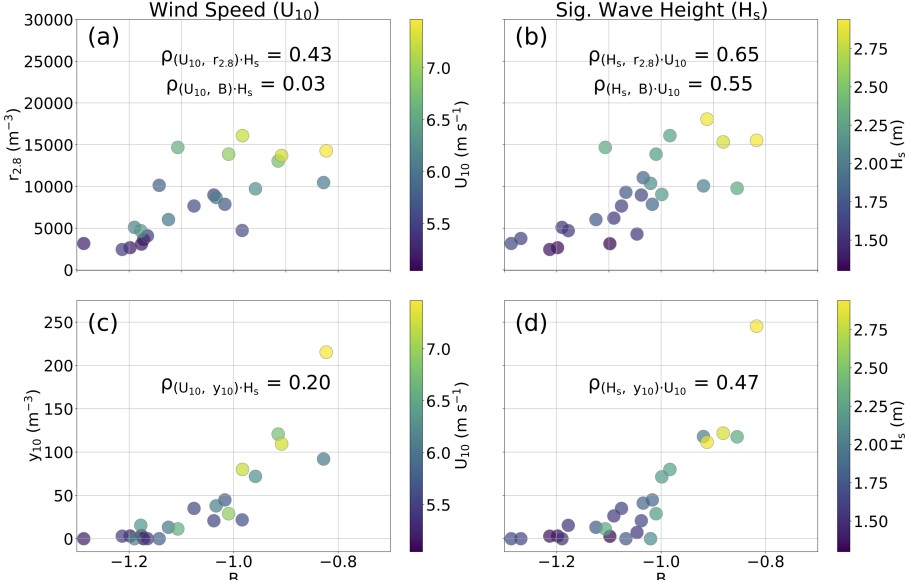

**Figure 8.** Three SSA-SD shape parameters are plotted: $r_{2.8}$, B, and $y_{10}$, which are binned and colored by $U_{10}$ and $H_s$. Plots (a) and (c) are binned by $U_{10}$, while plots (b) and (d) are binned by $H_s$. Shape parameter B is used for the x-axis of all plots. The partial correlations between these shape parameters and the environmental variables from Table 2 are also shown.

Another important aspect of a SSA-SD's correlation to the environment is the ability to accurately predict $y_{10}$. As seen in Sect. 3, COAST SSA-SDs show a larger increase in $y_{10}$ concentrations than smaller GSSPs when compared to OCEAN SSA-SDs hinting that production processes for ultragiant SSPs may differ from the coast to open-ocean. Capturing these changes in a fitted distribution can be difficult, though, because the shape parameters are more likely to be set by the smaller-sized, more plentiful SSPs. Therefore, analysis of $y_{10}$ to both $U_{10}$ and $H_s$ highlights whether SSA-SD shape parameters are accurately represent the ultragiant SSPs in this study. Figure 8c and d show the dependence of $y_{10}$ on B, colored by $U_{10}$ and $H_s$ respectively. Minimal differences in scatter distribution shapes emerge and both $U_{10}$ and $H_s$ have strong Pearson's r correlations to $y_{10}$ (Table 2).

Similar to Figure 8a and b, the visual correlation of $y_{10}$ to $U_{10}$ and $H_s$ (Fig. 8c and d, respectively) differs slightly, with $H_s$ appearing more organized than $U_{10}$. Partial correlations show that $H_s$ has the strongest correlation to $y_{10}$, but there remains a weak partial dependence of $y_{10}$ on $U_{10}$. Because ultragiant SSPs fall within the size range for spume droplets, it's possible this wind dependence may come from offshore spume droplet production in regions where $U_{10}$ exceeds 9 m $s^{-1}$. These ultragiant SSPs have significant fall velocities, and therefore $U_{10}$ may play an important role in modulating not only the maximum horizontal travel distance of these ultragiant SSPs, but also their vertical transport due to changes in turbulent kinetic energy.





**Table 3.** This table shows a collection of historical in-situ sea salt studies in coastal areas. The observed range of SSPs for each study has been converted to $r_d$ from the studies original range. If the ambient humidity wasn't considered in the particle size, the formation radius ($r_{98}$) or 80% RH was assumed based on the sampling altitude in the study. The measurement type (either total sea salt mass (Mass), total particle concentration (Concentration) or size distribution for the particle range (Size dist.) is recorded, along with which variable correlated most strongly to each study's observations (Sig. Var.) and the dates for which the study occurred (Duration).

| SSA Study | Particle Range ($r_d$) | Altitude Range | Measurement | Sig. Var. | Duration |
|---|---|---|---|---|---|
| Exton et al. (1985) | 0.05-12 $\mu$m | 10 m | Mass and Concentration | $U_{10}$ | Jun. 11-13 (1983) |
| Daniels (1989) | > 0.5 $\mu$m | 10 - 135 m | Mass | $U_{10}$ | 7 days between Nov. 2 (1981) - Mar. 8 (1982) |
| Petelski and Chomka (1996), Chomka and Petelski (1997) | Unknown | 0.5-6 m | Mass and Concentration | $H_s$ | Unknown |
| Piazzola and Despiau (1997) | 0.025-2.5 $\mu$m | 1-6 m | Concentration | $U_{10}$ | Dec. 12 - 14 (1994), Apr. 12 - 14 (1995) |
| De Leeuw et al. (2000) | 0.01-1.5 $\mu$m | 7 m - 15 m | Size dist. | $U_{10}$ | Jan. 24 - Feb. 5 (1996), Apr. 1 - Apr. 11 (1997) |
| Reid et al. (2001) | 0.5-6 $\mu$m | 0-400 m | Size dist. | $U_{10}$ | Feb. 26th - Mar. 12 (1999) |
| Clarke et al. (2006) | 0.005 - 5 $\mu$m | 20 m | Size dist. | $U_{10}$ | Apr. 21-30 (2000) |
| Van Eijk et al. (2011) | 0.025-1 $\mu$m | 6-16 m | Size dist. | Wave energy dissipation | Nov. 2 - 29 (2006), Oct. 16- Nov. 9 (2007) |
| Yang et al. (2019) | 0.05-3 $\mu$m | 11-18 m | Size dist. | $H_s$, Wave Reynolds # | Feb. 17 - Mar. 1 (2015), Dec. 21 (2016) - Feb. 16 (2017) |

## 5 Vertical Distributions of SSA-SDs

In-situ coastal studies have historically observed SSA within the bottom 30 m of the atmosphere (Table 3). To gain strong a quantitative understanding of SSA production, historical studies measured total mass, total particle concentration, or SSA-SDs to characterize SSA production at the air-sea interface (Table 3). Remote-sensing studies like Gathman and Smith (1997), De Leeuw et al. (2000), and Porter et al. (2003) observed that SSA production plumes can rapidly spread vertically to higher altitudes than observable by many in-situ studies, and that in-situ point-source measurements often missed a bulk of SSA production when compared to remote-sensing observations. While this study also utilizes in-situ point-source measurements, simultaneous sampling at multiple altitudes provides increased observational capacity of vertical mixing for different particle sizes as well as insight into how well upper altitude SSA-SDs correlate to changes in their local environment.





Total average mass for five altitude bins from 80 to 640 m were calculated for each sampling day, showing minimal variability
in total mass with altitude (Fig. 9a). The largest differences between samples, instead, come from the changes in environmental
conditions between sampling days. Overall, the three central altitude bins experience a decrease in total average mass with
altitude and a small decrease in total GSSP concentration, but there are several days in which total mass increases with height
from the lower altitude bins.

Masses and concentrations are difficult to interpret by themselves, though, as a wide range of B and $r_{2.8}$ combinations could
produce the SSA masses and concentrations observed in this study. Figure 9b shows the averaged SSA-SDs for the three central
altitude bins from sampling days with solid lines in Figure 9a. Despite the significant $v_\infty$ of ultragiant SSPs at upper altitudes
($\geq 0.2$ m s$^{-1}$, Fig. 4b), B, $r_{2.8}$, and total GSSP size range remain very similar with altitude and indicate that on average, the
MBL is very well mixed at our sampling location.

Lastly, two days from OCEAN and COAST with nearly identical $U_{10}$ and $H_s$ values were compared to evaluate whether
coastal dynamics influence SSA-SD changes with altitude (Fig. 10). Over the more limited altitude range, observations confirm
similar patterns from Sect. 3; the COAST concentrations m$^{-3}$ $\mu$m$^{-1}$ are greater than the OCEAN samples at all altitudes,
and the COAST samples observe a larger total range of GSSP sizes at each altitude than the OCEAN samples. Interestingly,
though, changes in the SSA-SD shape parameters are more significant. COAST $r_{2.8}$s increase with altitude while the OCEAN's
decrease steadily with altitude. Because these sampling days have similar environmental conditions, these results strengthen
the conclusion that the coast aids in increasing SSA production, but doesn't explain why COAST samples could experience an
increase in $r_{2.8}$s with altitude. We hypothesize that this increase is likely due to local dynamical controls on the vertical mixing
in that region, and that the fall velocities and production location in proximity to the coastline of these GSSPs may prove more
important.



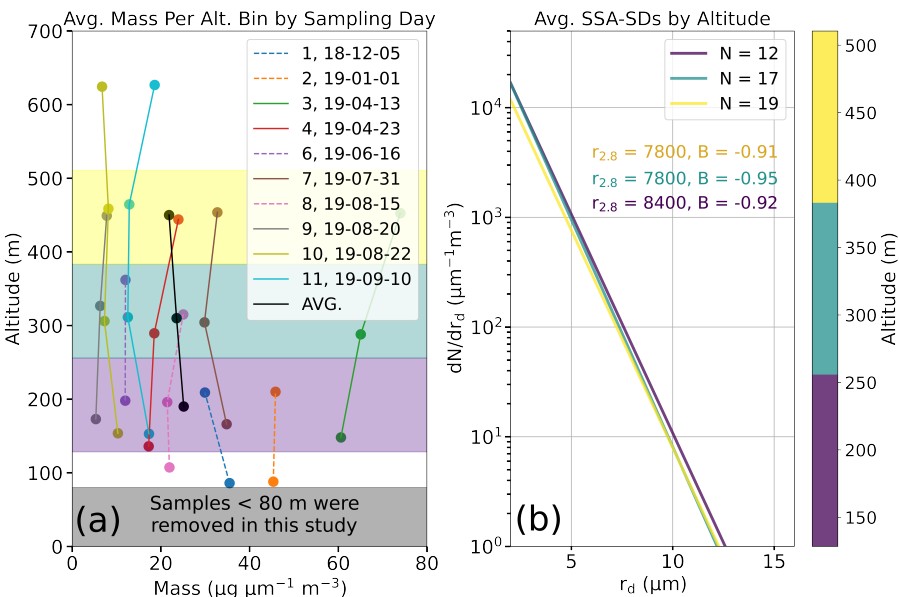

**Figure 9.** (a) Averaged total SSA mass for individual samples were plotted across five altitude bins: 80-128 m, 128 - 256 m, 256 - 383 m, 383 - 510 m, and > 510 m for individual sampling days. Within the three central altitude bins (128 - 256 m, 256 - 383 m, and 383 - 510 m), SSA masses were averaged across all sampling conditions to show how samples change with altitude (black solid line). Mass for the central bins decreased with altitude (5.03 $\mu$g m$^{-3}$, 4.71 $\mu$g m$^{-3}$, and the highest bin 4.37 $\mu$g m$^{-3}$). (b) The samples from solid lines in (a) were averaged together to create SSA-SDs for the three central altitude bins. These depict how SSA-SDs change with altitude for three sample days rather than only mass.





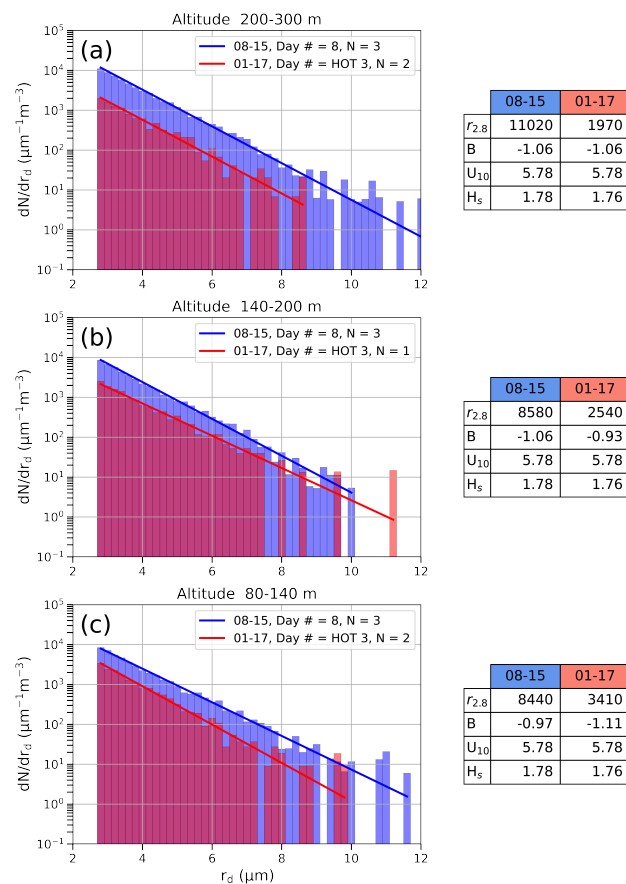

**Figure 10.** Two sampling dates from the OCEAN (red) and COAST (blue) with similar environmental conditions were chosen to compare changes in SSA-SDs with altitude. Three discrete altitude bins were compared based on the availability of samples between these two days. SSA-SD parameters $r_{2.8}$ and B, along with the two environmental variables $U_{10}$ and $H_s$ are shown in tables to the right of each graph.



## 6    Modeling Coastal Orographic Impacts on SSA-SDs

While we've investigated the changes to SSA-SDs between the coast and open-ocean, it's difficult to remove the influence of the open-ocean on these observations. More plainly, these coastal observations likely contain a mixture of GSSPs produced in the open-ocean and the coastal region. Therefore, SSA trajectories help to supplement how the production location and environment wind speeds affect different sized GSSPs' vertical mixing potentials.

Fall velocities from Figure 4b show that a singular particle's $v_\infty$ can vary by an order of magnitude from the ocean surface
to cloud base. The smaller GSSPs in this study have negligible fall velocities (Fig. 4b) and can remain in the atmosphere for days to weeks, meaning their ability to travel freely through the atmosphere is limited mainly by wet deposition (Lewis and Schwartz 2004, p. 76). The larger GSSPs can remain suspended for an average of just minutes, meaning their ability to be vertically and horizontally mixed is more limited by their masses (Lewis and Schwartz 2004, p. 76). As a result, larger GSSPs and ultragiant GSSPs require stronger updrafts to move similar horizontal or vertical heights as their smaller counterparts. This
is an overall limiting factor of atmospheric concentrations of ultragiant SSPs.

Trajectory ranges for three radii of GSSPs are shown for three production locations for the high and low wind days (Fig. 11). When particles were released at the Coastline production location (100 m offshore), all three GSSP radii were vertically mixed regardless of their size (Fig. 11a, d, g). A small difference in average maximum altitude obtained in this domain (AMA) is observed between $r_{d2.8}$ and $r_{d12.2}$ for both wind simulations, and the low wind day sees smaller AMAs across all GSSP sizes.
No GSSP size experienced dry deposition, indicating that particles released in the Coastline become rapidly vertically mixed by the coastal orographic updrafts on both wind days.

At the Intermediate production location (2600 m offshore), all GSSPs experienced increased AMA except for $r_{d12.2}$ on the lowest wind day. The AMA for $r_{d12.2}$ decreased by almost 40% (Fig. 11h) for the low wind simulation, whereas $r_{d2.8}$ and $r_{d7.4}$ increased between 11%-22% (Fig. 11b, e, h). The total altitude range of the $r_{d2.8}$ particle tightened compared to the Coastline
production location, indicating that smaller particles released in the Intermediate location experience smaller variability in their trajectories compared to the $r_{d7.4}$ and $r_{d12.2}$ particles, likely due to the minimal fall velocity of this GSSP. The Intermediate production location begins to demonstrate some of limitations created by the production location for the larger particles. The fall velocity of the $r_{d12.2}$ means some trajectories experienced dry deposition, fewer $r_{d12.2}$ particles become vertically mixed, and the AMA was smaller overall when compared to the Coastline location.

The Open-Ocean production location (5800 m offshore) strongly alters the trajectory ranges and AMAs for all particles compared to the Coastline location. The $r_{d12.2}$ AMA for both wind simulations is significantly reduced, with no $r_{d12.2}$ particles reaching the coast on the low wind day. The $r_{d7.4}$ particles also become distance limited for the low wind day, with an approximate 50% decrease in AMA. The $r_{d2.8}$ particle, though, continues with similar trends at the Intermediate location; an increased AMA for both wind days, as well as a tightening of the trajectory range. It is likely that $r_{d12.2}$ $v_\infty$ is approximately equal to
or greater than the average updraft speeds over the open-ocean surface, significantly decreasing the likelihood these ultragiant SSPs become vertically mixed when produced at large distance from the coastline (> 6 km). For many of these ultragiant SSPs,





their average atmospheric lifetime is less than 10 minutes (Lewis and Schwartz 2004, p. 77), meaning they unlikely to travel more than 6 km from their origin when horizontal winds are less than 10 m s$^{-1}$.

These trajectory ranges offer a small glimpse into how wind speed and production location can affect the SSA-SDs observed
in this study. Interestingly, r$_{d2.8}$ particles released closer to the coastline experience smaller AMA and a wider total altitude range than r$_{d2.8}$ particles released farther away. For our study, this may indicate that COAST samples taken at higher altitudes observe r$_{d2.8}$ particles produced anywhere from the Coastline through the Open-Ocean, a large overall production area. COAST samples taken at lower altitudes are more likely to observe r$_{d2.8}$ particles only produced near the coastline, a smaller production area for r$_{d2.8}$ particles observed at this altitude. Therefore, as COAST samples increase in altitude, the production area increases
to greater distances away from the coastline for r$_{d2.8}$ particles, resulting in higher concentrations of these small particles relative to the lower altitude samples.

To the opposite effect, ultragiant SSPs like r$_{d12.2}$ experience increased rates of dry deposition in weak open-ocean updrafts, meaning their production areas are strongly distance-limited. Therefore, ultragiant SSPs concentrations at higher altitudes were likely produced in the Coastline or nearshore environment, whereas ultragiant SSP concentrations at lower altitudes may come
from a larger production area resulting in a decrease in ultragiant SSP concentrations with altitude. In these simulations, we did not observe major limitations caused by changes in orographic updraft velocities between the high and low wind days to any GSSP sizes, indicating that orographic updraft speeds for moderate trade wind days are strong enough to vertically mix all SSP sizes in this study. Instead, the limitation on a particle's vertical mixing potential was dominated by the production location. These effects likely play a role in skewing SSA-SD shape parameters, but a more robust set of observations and modeling
capabilities should be explored to further improve our understanding of coastal effects on SSA-SDs.




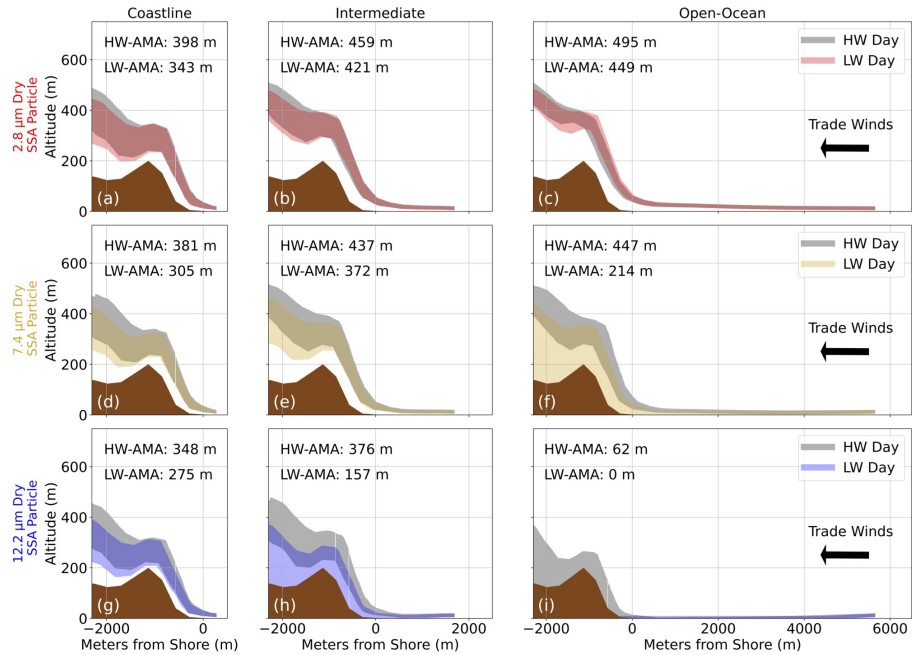

**Figure 11.** Cross sections from Figure 3 were utilized to show trajectory ranges for both the high and low wind days across the three production locations. Three different sized GSSPs were released at three production locations within these cross sections. (a), (b), and (c) show trajectories for $r_{d2.8}$, (d), (e), and (f) show trajectories for $r_{d7.4}$, and (g), (h), and (i) show trajectories for $r_{d12.2}$. Colored regions remain the same within particle sizes and show trajectories ranges for the low wind day (LW), while all grey regions show trajectories ranges for the high wind day (HW). The HW and LW average maximum altitude (AMA) are given for each plot, and the trade wind direction is generally represented as right to left.



## 7 Discussions and Implications

### 7.1 Important Considerations for Environmental Variables

#### 7.1.1 Significant Wave Heights vs. Wind Speeds

In this coastal study, $H_s$ has a stronger correlation to SSA shape parameters than $U_{10}$. There does appear to be a strong
correlation between $U_{10}$ and $H_s$ (Pearson's r = 0.85) for our samples, though, which could imply that $U_{10}$ is still a viable proxy
to represent coastal SSA production. Further investigation into this relationship, however, shows that the correlation between
$U_{10}$ and $H_s$ across an extended climatology is only Pearson's r = 0.55 for this area, and therefore the correlation we observe
between these two environmental variables is artificially high. The strong correlation in our study is likely because we require
onshore trade winds to sample and therefore are more likely to observe days with trade wind swells. In the context of historical
coastal SSA studies, it's possible that sampling restrictions have resulted in an artificially strong correlation of SSA-SDs to
$U_{10}$, too. Therefore, it's possible that utilizing $U_{10}$ in a coastal production equation for this region may misrepresent the actual
production.

Once the co-relationship between $H_s$ and $U_{10}$ is removed, partial correlations confirm that $H_s$ is the only significant variable
to setting coastal SSA-SD shape parameters. Utilizing $H_s$ not only makes sense from a hydrodynamical perspective (Chomka
and Petelski 1997), but has the potential to reduce error in production estimates created by anomalous wind estimates. In the
analysis by Grythe et al. (2014), many open-ocean SSA production equations had excessive generation of spume droplets due
to anomalous high $U_{10}$ events, biasing global production estimates to be larger than observations. $H_s$ often has smaller standard
deviations ($\sigma$) than $U_{10}$, and therefore replacing $U_{10}$ with $H_s$ as the primary variable to coastal SSA production may minimize
the sensitivity of the production equation. It should be noted, however, that $U_{10}$ should not be completly removed from coastal
SSA equations, as it is essential for spume droplet production when $U_{10} > 9 \text{ m s}^{-1}$ (De Leeuw et al. 2000).

#### 7.1.2 Considerations for Sea Surface Temperature

Another environmental variable that was analyzed but not previously discussed is SST. Typically one would expect increasing
SSA production with increasing SSTs, as found in Anguelova and Webster (2006) and Jaeglé et al. (2011). Correlations between
SST and SSA-SDs in this study, however, show the opposite. A moderately negative correlation was found between the SSA-
SDs and SST, but we hypothesize this was due to our specific sampling environment and wave behavior experienced in Hawaii.
There is a strong seasonality of both SST and $H_s$ in Hawaii. During the boreal winter, the northern Pacific Ocean generates large
waves during strong Aleutian Lows that propagate southwards towards the Hawaiian Islands. As a result, the northern shores
experience peak $H_s$ values when SSTs are the lowest in Hawai'i. In the boreal summer, storms in the southern Pacific generate
waves that travel northwards towards the islands, meaning large waves reach the southern shores when so when SSTs are the
highest. This study took place on an eastern coastline across a one year period, meaning our largest $H_s$ values came during the
boreal winter when Hawaii experiences the lowest SST. While our observed range of SST is relatively small (approximately
3.5 °C), we anticipate that $H_s$ plays a much larger role in SSA-SD variability than SST in this location.



## 7.2 Considerations for Future Observations of Coastal SSA

### 7.2.1 The Significance of the Aerosol Size Distributions

Gathering in-situ observations of SSA-SDs is notoriously difficult. Sampling limitations are imposed by size and weight of aerosol particle sizers, accessibility to aircraft, and instrumentation costs, resulting in reduced sampling locations and frequency. Additionally, many studies characterize SSA changes based on the total mass or concentration rather than changes in size distributions due to the nature of these aerosol samplers. The lack of complete SSA-SDs has resulted in the implementation of dynamic modeling approximations Porter and Clarke (1997); Blanchard et al. (1984), in which SSA size distributions are

assigned based on concentrations alone, independent of the environment conditions.

Section 4 demonstrates how SSA-SDs shape parameters can vary across different $H_s$ and $U_{10}$, and therefore assigning distribution shapes blindly may be unphysical (Reid et al. 2001). Section 5 shows how SSA-SD shapes change between the open-ocean and coastline, implying that open-ocean SSA-SDs may not be suitable to apply to coastal observations. Lastly, Sect. 6 shows how different altitudes can have different sources of SSPs; upper altitudes are more likely to see higher concentrations

of smaller-sized GSSPs from the Open-Ocean, Intermediate, and Coastline regions, while lower altitude samples are more likely to contain ultragiant SSPs from these regions. Despite GSSA existing in such small concentrations, they can drastically skew a SSA-SD shape depending on the sampling altitudes and the sources and sizes of SSPs present. For these reasons, observing SSA-SDs is important.

### 7.2.2 Observations of Giant Sea Salt Particles

A unique aspect to our sampling methodology is the ability to observe GSSPs. Because these GSSPs exist in smaller concentrations than submicron SSPs, larger sampling volumes are necessary to observe them. Additionally, their size, and consequently their inertia, means aerosol inlets experience larger rates of particle loss for GSSPs, making samplers with inlets inadequate for studying these larger size ranges. Table 2 from (Grythe et al. 2014) shows that only three of the 22 analyzed source functions account for ultragiant SSPs and half of the equations only accounting for $r_d < 5$ $\mu$m, while coastal samples from this study's

Table 3 only has 2 studies that can observe $r_d < 5$ $\mu$m.

Despite GSSP concentrations being much smaller than submicron SSP concentrations, they remain significant in terms of their particle mass and energy transfers, as well as their potential to act as rain-drop embryos and GCCN in clouds (Lasher-trapp et al. 2005; Cooper et al. 2013; Jensen and Nugent 2017). Observing a complete range of particle sizes within a size distribution may prove vital to bettering estimates of global SSA mass and volume. We hope that the affordability and accessibility of the

mini-GNI instrument may encourage future studies to consider sampling GSSPs in addition to their original SSP size range.

### 7.2.3 Accounting for Local Dynamical Controls

Few studies have observed changes to SSA-SDs along the coastline, and when available, most observations occur within the lowest 20 m of the atmosphere (Table 3). SSA plume heights are highly variable and depend on local conditions like coastal





wave energetics (Chomka and Petelski 1997) and wind speeds (De Leeuw et al. 2000). Additionally, the resulting SSPs can

experience rapid vertical mixing from their production location by coastal thermals (Porter et al. 2003) and orographic lifting. If the in-situ sampling method is altitude restricted, these studies may not observe the full production potential within this location, and may be more likely to see specific SSP sizes based on conclusions in Sect. 6. GSSPs and ultragiant SSPs are more likely to experience fall-out when they're produced at great distances from the coast, generating steeper gradients for these particles with altitude. Smaller GSSPs and submicron SSPs may remain aloft longer, meaning observations in coastal

environments may be unfairly attributing the increase in smaller SSP concentrations to coastal production if they do not consider the contributions from the open-ocean at various altitudes.

## 8 Conclusions

This paper establishes important precedents for coastal SSA studies in the future, as well as highlighting the significance of $H_s$ on changes to SSA-SDs in coastal environments. We set out to answer three questions:

1) How do SSA-SDs change from the open-ocean to the coastal environment?,

    2) What environmental variable is most significantly correlated to changes in SSA-SDs? $U_{10}$ or $H_s$?, and

    3) How do dynamics within our sampling location contribute to variation the SSA-SDs across different environmental conditions and altitudes?

There are notable differences between observations of SSA over the open-ocean and the coastline. SSA-SDs change in

cumulative concentration as well as distribution shape between these regions, indicating there may be more complicated processes in the coastal environment with regards to the production of GSSPs and ultrgiant SSPs. For example, coastal SSA-SDs on average have gentler slopes, meaning there exists increased concentrations of ultragiant SSPs compared to the open-ocean. When the COAST observations were compared to historical cumulative concentrations from Woodcock (1953), OCEAN $N_c$s were almost identical to W53, while the COAST $N_c$s were elevated across all observed wind and altitude ranges.

Analysis of environmental changes on coastal SSA-SDs shows a clear dependence on $H_s$. The visual correlation of SSA-SDs to $H_s$ is stronger than $U_{10}$, and the partial correlations confirm the significance of $H_s$ for SSA-SD shape parameters B and $r_{2.8}$. Partial correlations also confirmed that the strong Pearson's r and Spearman's $\rho$ between $U_{10}$ and the coastal SSA-SD shape parameters came from the strong correlation between $U_{10}$ and $H_s$ in this study. It is important to note, however, that the correlation between $U_{10}$ and $H_s$ is stronger in this study than in nature, due to restricting sampling to days with trade wind

conditions. This is an important takeaway for future studies of coastal SSA-SDs, as sampling under specific environmental conditions may result in artificially high correlations.

Altitude observations and models highlighted the important roles that production distance from the coastline, $U_{10}$, and coastal orography play in setting SSA-SDs. For smaller GSSPs, fall velocities are negligible, meaning these particles are unaffected by their production distance from the coastline and have strong vertical mixing potential. Larger, ultragiant SSPs

have significantly shorter atmospheric lifetimes and the concentrations observed in this study are significantly limited by their production distance from the coastline. Because of this, observations of SSA in a coastal environment for smaller GSSPs are





likely to see a higher combination of open-ocean and coastally produced particles, while larger and ultragiant GSSPs are likely to have been produced much closer to the coastline.

Overall, these conclusions are important considerations in future campaigns that observe coastal SSA production. The analyses of coastal features specific to this sampling location provide significant insight into how they modify our SSA observations, and set a precedent for future discussion of coastal SSA production. These results could have implications for modeling, as the coastline has been shown to produce higher concentrations of ultragiant SSPs compared to the open-ocean, and orographic coastlines provide significant lift for GSSPs and ultragiant SSPs not typically seen over the open-ocean.



*Author contributions.* ADN provided the funding acquisition for the study and project administration. CT, ADN, and KLA participated in the
conceptualization and data curation in this study. CT provided early code and environmental data for the formal analysis and methodology.
KLA created the methodology for the WRF and trajectory model, executed and formally analyzed the results for the models and in-situ
samples. KLA and ADN contributed to the visualization of this paper, and KLA wrote the original draft. KLA and ADN contributed to
review and editing.

*Competing interests.* The authors declare no conflicts of interest relevant to this study.

*Acknowledgements.* This work was supported by funding through NSF AGS Award 1762166 with the title "EAGER: A New Method for
Sampling Sea Salt Aerosols" (Nugent), as well as NSF CAREER Award 2145502 with the title "Quantifying the Sea Salt Aerosol Size
Distribution in the Coastal Atmosphere: The Role of Wind and Waves" (Nugent). Additionally, the authors thank Jørgen B. Jensen, whose
guidance, experience, and expertise were essential contributions to the gathering and analysis of the sea salt aerosol data.



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
