# Peer review of "Mechanisms controlling giant sea salt aerosol size distributions along a tropical orographic coastline"

_EGUsphere, 2023_

## Author Comment (AC1)

**Response to Reviewer #1**

By Katherine L. Ackerman, Alison D. Nugent, and Chung Taing

The manuscript "Mechanisms controlling giant sea salt aerosol size distributions along a tropical orographic coastline" presents measurements of sea salt aerosols, including giant and ultragiant particles, from a novel device both on the coast and in the open ocean. Notable differences in the overall SSA concentrations are noted between the two sites, as are differences in the SSA-SD shape parameters and vertical profiles. Some of the mechanisms are illustrated by trajectory tracking in a WRF run of the area.

In my opinion, this is a great study and a very well-written manuscript. It is well-referenced and clear, and a lot of details were attended to properly. It also provides very reliable measurements of a data-scarce topic: SSA and giant particles in particular, but also comparing coastal to open ocean conditions. A lot remains to be understood about how universal the SSA-SD shape parameters are, and this study does the field a service by so clearly demonstrating some of the key differences that can occur. I only have a few very minor points for the authors to consider, but otherwise think that this would make a great contribution to ACP.

We would like to thank the reviewer very much for their thorough review of our paper! Thier recommendations have helped to clarify key details in the study and strengthen the major points we were hoping to emphasize. We appreciate the time and effort put in and are grateful to have received constructive and encouraging comments. Our responses to the specific comments are in detail below:

1. Line 9: Maybe should be "facilitating"?
   Thank you very much! It has changed from facilitate to facilitating.

2. Sentence starting on line 45: Maybe consider "whitecap fraction" explicitly as one of the things that has been analyzed a lot? It seems to fit with the way the literature has been broken up here – production has been linked with those things listed (U10, SST, SSS) but also whitecap fraction.
   We agree – this sentence was broken up and restructured for clarity to indicate that U10 is typically used to directly and indirectly represent SSA production (by increasing whitecap fraction). The paragraph now reads:

   "To characterize these whitecap interactions, in-situ, remote sensing, and laboratory experiments analyzed how environmental parameters such as 10 meter wind speeds (U10) contribute to the production of SSA, either as a direct mechanism for generating spume-sized SSP or as an indirect mechanism by increasing the whitecap fraction (De Leeuw 1986; Monahan et al. 1986; Andreas 1998; Gong 2003; Lewis and Schwartz 2004; Clarke et al. 2006; Petelski and Piskozub 2006; Andreas et al. 2008; Norris et al. 2008). Other studies have built on these findings by including additional ocean surface characteristics like sea surface temperature (SST) (Mårtensson et al. 2003; Jaeglé et al. 2011; Zinke et al. 2022), and sea surface salinity (Sofiev et al. 2011; Zinke et al. 2022)."

3. Figure 2: It might be nice to have a legend in the figure itself, explaining the symbols (so the reader doesn't have to sift through the caption)
We agree, we've added a legend to the top right corner.

[Figure]

4. Line 177: I'm not sure what the "Bellows' 12 m" wind speed is here. I searched around the manuscript but didn't see any description of what this meant. So I'm left a little uncertain of why it was regressed to the WeatherFlow, and which one was ultimately used for the reported U10 values
Bellows was indeed not previously defined, the sentences in these paragraph now read as:

"A linear regression on almost two years' worth of data was conducted between a 12 m tall anemometer approximately 7 km northwest of Kaupō Bay (Bellows' 12 m) and WeatherFlow's 10 m anemometer for trade wind days (wind directions between 30-90 degrees). The Pearson's r between these two locations was 0.904 and significantly better correlated than our in-situ 2 m Kestrel samples to Bellows' 12 m wind speeds (Pearson's r = 0.681). Therefore, a relationship between the WeatherFlow 10 m wind data and the Bellows' 12 m wind data was derived to approximate historical wind speeds for the SSA sampling dates."

5. Line 206: What is meant by "the Shin and Hong PBL scheme for the grid resolution"? Is the phrase "for the grid resolution" supposed to be there?
Thank you for catching this! "For the grid resolution" was removed for clarity, and PBL was expanded to say planetary boundary layer.

6. Line 231: Are these using the averaged wind profiles? Or the time-varying profiles from some set frequency of WRF output? It appears to be the average, so it's probably worth stating explicitly here.
Thank you for asking this clarification question – yes, this is the 3-hour averaged wind

profile around 2 pm (the average sampling time across our studies) for both the low and high wind days. While we could have had time-varying profiles for the trajectories, we wanted these trajectories to represent an average circumstance for particle paths in high wind vs. low wind circumstances. We've updated this section so that our intentions are clearer to the reader:

"Lastly, trajectories for GSSPs sized $r_{d2.8}$, $r_{d7.4}$, and $r_{d12.2}$ were simulated using the 3-hourly averaged WRF wind profiles around the average time for our in-situ samples (2 pm local time) as well as the calculated fall velocities for both wind days."

7. Line 245: This isn't necessarily a suggestion for this study, but in the future it might be more realistic to use the PBL scheme parameters to approximate this variability.
   This is an excellent suggestion for the future! We'd love to explore how these trajectories change under more realistic conditions once we gather a larger collection of samples.

8. Line 267-269: If I'm reading figure 6 correctly, this sentence is really only referring to figure 6b (at specific heights), right? The COAST samples in 6a don't always exceed W53 at the wind speeds mentioned in this sentence. However I might be misunderstanding because the next paragraph explicitly introduces fig 6b.

[Figure]

We have updated this figure to be more colorblind friendly, as well as with additional annotations to better highlight some of the subtle differences. At first it might appear that only the smaller wind range COAST Nc (the solid navy line) exceeds the W53 sample (dashed navy line). We have highlighted the larger differences between the COAST and W53 Ncs for the next wind range in figure 6a. For this next range, the COAST Ncs are from winds between 5.5-7.9 m s$^{-1}$, while the W53 Ncs are from wind speeds from the range 8.0-10.4 m s$^{-1}$. So while they appear to be extremely close, the W53 samples are from a higher wind speed range, and we therefore assume that if our wind speed range had been within this range our COAST Ncs would exceed the W53 Ncs. We realize, though, that the green dashed line is used twice between Figure 6a and 6b for W53 samples even though the wind speed range differs between these two figures.

9. Line 314: "chances" should be "changes".
   Thank you for catching this error, this has been corrected.

10. Line 315: "strong" should be "stronger".
Thank you for clarifying this sentence - "remains strong" was changed to "becomes stronger".

11. Line 351: I'm not sure the units are needed after "concentrations"
These units were removed as they were explained in previous sections of the paper.

12: Figure 4: It would be helpful if this figure had a panel "c" which showed the average RH profile that these are based on.

[Figure]

Thank you for this suggestion! To save space, we added the average RH profile for the August sampling date to both a and b plots. Now, readers can compare how relative humidity affects the deliquesced particle size of the different dry radius particles, and consequently the fall velocities.

13: Line 410 (or thereabouts): Somewhere at the end of this section I think it's important to point out the relatively simple way of doing the trajectory simulations. I think it's totally appropriate for what the authors are trying to do, but the treatment of turbulence in particular is only very crudely represented. I think the authors do a good job of only drawing conclusions which are supported by this method – I just think that a brief 1-sentence reminder at the end of the processes (especially turbulence) which were left out and could lead to significant differences in some of the details.

Thank you very much for this suggestion! We definitely want to emphasize that these results do not definitively represent all coastal processes, but merely help to demonstrate how production distance could play a role in changing the concentrations of different sized sea salt particles at different altitudes. To make this clearer, the following two sentences have been amended to say:

"These simulations offer a simplified representation of coastal orographic processes and further investigation into the roles of turbulence and coastal controls on dynamics should be completed, but overall, they demonstrate the potential impact that distance plays in controlling coastal SSA-SD shape parameters. Future studies will likely require a more robust set of observations and modeling capabilities to improve our understanding of coastal dynamics on SSA-SDs."

14: Lines 435-436: Elsewhere it's spelled "Hawai'i"
Thank you for spotting this! This has been corrected.

---

## Author Comment (AC2)

**Response to Reviewer #2**

By Katherine L. Ackerman, Alison D. Nugent, and Chung Taing

This manuscript presents the results of a sea salt aerosol (SSA) study in coastal zone (Hawaii) at moderate wind speeds (Trade winds up to 7 m/s). Super-micron SSA size distributions (dry radii > 2.8 mm) were measured with a miniature impaction instrument, mini-GNI, specifically designed for these observations. The size distributions in coastal zone were compared to SSA size distributions in open ocean and differences up to a factor of 5 are reported. Several mini-GNIs were deployed on a kite to obtain the vertical profiles of SSA number concentrations from 80 to 650 m height. Various environmental data were collected in situ (wind, wave height, air temperature, relative humidity) to identify the most suitable forcing variables for the observed SSA size distributions. Data generated with WRF model were used to simulate SSA trajectories and study the effects of the coastal topography on the SSA vertical dynamics.

Super-micron size distributions of SSA, as well as their vertical profiles, in coastal zone are indeed severely understudied. Though fewer and less frequently occurring, large SSA particles, with their large areas and volumes, contribute disproportionately to various aerosol effects, from scattering and absorption to formation of cloud condensation nuclei. The differences between open ocean and coastal zone conditions yield dramatic variations of SSA production, yet the governing processes are not well characterized. Studies like the one presented here are sorely missing and these new results represent a great new addition of data on SSA production and mixing as well as on processes affecting them.

This is a very well planned and executed study involving a myriad of observations and data. The manuscript is well organized and written. Considering the amount of used instrumentation, modeling, data collection, and analyses, the manuscript is concise and clear. The results are novel and represent valuable contribution to the representation and understanding of air-sea interaction processes. I recommend this manuscript for publication at ACP. I have minor questions and remarks. These are listed below by line number, not importance.

We extend our heartfelt gratitude to the reviewer for the thorough review of our paper, as well as words of encouragement for this study. Their meticulous analysis has significantly enhanced the readability and conclusions of our paper. The reviewer's praise is encouraging for further work in this field, and we're grateful for the time and efforted invested by the reviewer in evaluating our paper and offering suggestions. We look forward to implementing their suggestions to further elevate the quality of our paper. The specific comments are addressed below:

1. Line 23: Suggest change "SSP" to "sea salt particles" (thus avoid defining an acronym that is not further used in the abstract)
   Thank you for catching this! We've changed SSP to sea salt particle.

2. Lines 30 and 32: Acronyms SSA and SSP seem to be used interchangeably throughout the text. If so, better use "SSA" and "SSA particles" for consistency. If they are meant to represent some differences, it would be good to mention those here in lines 30-31 on first encounter.

Thank you for your careful attention to this detail. Literature surrounding SSA has been largely inconsistent in the reference to the aerosol species in general vs. the physical particles in the atmosphere, and our intention was to be more discrete in this manuscript. After re-reading the introduction, however, we agree that our use of SSA vs. SSP was inconsistent. We have updated these sections and carefully re-read the consequent sections to ensure that our use of SSA remains as a broad reference to the aerosol species (which encompasses a broad range of categories like measurement types, chemical properties, or other factors describing the aerosol species), while the use of SSP is reserved for discussion of the observation, quantification, and properties of the particles themselves.

3. Line 31: Symbol $r_d$ is used with both italic and non-italic (e.g., line ) subscript throughout the text and in the equations. Use one notation consistently. All other variables are non-italic. That is fine. Just check the ACP requirements for math/variables style.
   Thank you for noticing this – we've updated the text so that, with the exception of the Greek letters, the variables are consistently non-italicized to match each other and the figures.

4. Line 34: "orders of magnitude less often"—Is this to say that the large particles are fewer in number and occur less frequently? Or is this phrase strictly for the frequency of occurrence?
   Thank you for the clarification question – we've changed the sentence as follows:
   "These giant SSPs (GSSPs, $r_d > 0.5$ µm) exist in atmospheric concentrations that are orders of magnitude less than their smaller counterparts, however."

5. Line 68: Remove "the" to read "winds below it act to dilute the SSA concentration"
   The section now reads "where winds above this threshold add to the SSA concentrations and winds below it act to dilute the SSA concentration."

6. Line 96: Remove comma before "sea state"
   This change has been made.

7. Line 112: Remove "sea salt aerosol size distribution" because acronym SSA-SD is already defined in line 91.
   This change has been addressed.

8. Line 133: Introduce the concentration units in parentheses here on first definition of N and do not repeat them anymore; i.e., no need to include units in lines 134, 257, 258, 351, etc) unless they follow a specific number.
   Thank you for this helpful clarification! We've changed the appropriate sentences so that the variable name and units are defined together only once.

9. Line 133: Unless ACP writing style allows, you should not start a sentence with a symbol, like here starting with $r_{2.8}$ and in line 135 with $r_d$. There are several other cases (e.g., line 255). Likewise, should not start a sentence with a number as in line 235.
   Thank you, the text has been worked around to ensure that sentences do not start with variables.

10. Line 136: After introducing $r_{2.8}$ and B in lines 131-135, it would be good to specifically say here that you call these shape parameters and also give their physical meaning, besides their definition.

Thank you for pointing this out – we want these equations to feel connected to the actual size distributions, as we refer to these variables quite a bit in the future sections. We've reworked the following paragraph for clarity:

"where $dN(r_d /dr_d)$ is the concentration (N, $m^{-3}$ $\mu m^{-1}$) at the bin (width 0.2 $\mu m$) centered on the dry particle radius ($r_d$). The number concentration ($r_{2.8}$, $m^{-3} \mu m^{-1}$) is for a dry particle with a radius of 2.8 $\mu m$ which is the smallest observable SSP size in this study. Because the function is fit across a range of dry particles valid for SSPs of $r_d >= 2.8$ $\mu m$ and visualized on a lognormal Y-axis, $r_{2.8}$ represents the Y-intercept for a given SSA-SD. The second changing parameter is B, which is the inverse of the characteristic radius for the size distribution with $B > 0$. The lognormal Y-axis represents the exponential decay equation into a linear line, meaning B represents the slope of a SSA-SD on a lognormal Y-Axis.

Both $r_{2.8}$ and B become the two SSA-SD shape parameters that change between SSA-SDs for different altitudes and environmental conditions and are analyzed further in this study. The inclusion of the -2.8 $\mu m$ in the exponent is implicit in the remainder of the equations used in this study due to the sampling range limitations. This function was fit to each size distribution from 2.8 $\mu m$ through the largest sized particle for continuous bins; i.e., the function stops fitting to particles when there was a break in observed SSP bins (Fig. 1b, red line). These exponential fits are then used in this study to represent the SSA-SDs for given samples if the $r^2$ values of the fits were $> 0.90$."

11. Table 1 caption: Change to $H_s$ (m) and Per (s) for consistency with the listing of all other variables. Case 5 is missing in column "Sample #"; perhaps it has been discarded due to CE < 40% or some other reason. It would be good to mention it to avoid impression of typo on the order of Sample #.

Thank you for these suggestions! Significant wave height was replaced with $H_s$, however, Period remained as "mean wave period (Per., s) because this variable was not abbreviated elsewhere in the text.

A sentence was added in the table caption for clarity that Sample #5 was removed from this analysis because the altitudes of the samples were not > 80 m.

"Sampling date #5 was not included in this analysis as the altitudes of all the samples were < 80 m."

12. Line 181: Altitude z is given in italic and non-italic. Here and everywhere, please use the same notation.

Thank you for noticing this, the remainder of the paper has been updated for non-italic values for everything except Greek letters.

13. Line 202: Need to define what is "h-values"
   Thank you for noticing this – "pressure derived height values" was added after eta-values in the text.

14. Lines 205-209: Acronyms RRTMG, PBL, EPSSM, and NCAR are not defined. They are not used anywhere else. Better not to use acronyms in this case.
   Thank you for these suggestions – the acronyms were removed and replaced with their original meanings.

15. Line 253: "Cumulative" seems to mean averaged over a range of altitudes and a range of winds speeds. If so, better say this specifically to avoid any confusion with the statistical meaning of "cumulative."
   Thank you for noticing this! We have changed this section title to be "Averaged SSA-SDs in the Coast vs. Open Ocean", as the figure this section is referencing is a SSA-SDs rather than a cumulative concentration plot.

16. Line 262: Short-hand W53 is not introduced. Good place is line 160 regarding historical samples; introduce it as "Woodcock (1953, W53)"
   Thank you for this suggestion, however, W53 is defined in section "2.2.2 Historical Samples" on line 166 of the original draft.

17. Figure 7 caption: Put the units for $U_{10}$ and $H_s$ in parentheses.
   Thank you for noticing this. Because $U_{10}$ and $H_s$ are already defined, we chose to remove the long names for the variables and the units put in parentheses. The sentence now reads: "SSA-SDs were binned and averaged by two environmental parameters - (a) $U_{10}$ (m s$^{-1}$) and (b) $H_s$ (m) - to visualize the organization of these size distributions to changes in the wind and waves"

18. Figure 9 caption: How were these altitude bins chosen? If the criterion for the altitude bin formation changed, would that affect the shape of the vertical profiles?

This is a great question that doesn't appear well clarified in our current manuscript. Throughout sampling, we explored a variety of altitudes and methodologies, resulting in a wide distribution of altitudes across our many sampling days, however, a general sampling strategy was to try for at least three different altitudes approximately 100-150 m apart. As a result, our distribution of samples across altitudes is a trimodal distribution, with the three modes around approximately 150 m, 300 m, and 450 m. The initial decision for choosing bins were to maintain a majority of the samples within three evenly-spaced bins. Additionally, we only chose days that had samples in each bin, as to not skew bins from days without a complete representation of altitudes for those environmental conditions.

Below is an additional plot showing our raw distribution of samples across altitude bins, the current distribution in the previous manuscript, and a new distribution encompassing a slightly different range between the three bins. The old distribution had altitude bins between 80 m, 128.4 m, 255.8 m, 383.2 m, 510.6 m and 638 m. The new distribution is not uniform, with bins from 80 m, 200 m, 360 m, and 510 m, but encompasses all samples except for those > 600 m and has a more even sample count between the bins. We have excluded the highest-level samples as they are too far away from the remaining distribution of samples and we do not have similar sample counts for a fourth bin.

[Figure]

[Figure]

Overall, changing these bins does not change our conclusions, but the reviewer makes an excellent point about the bins not being obvious. Therefore, we updated Figure 9 with the new, more inclusive distribution. The text in this section has been modified to account for the changes in bin ranges and reference to the physical changes in the figure, but the conclusions remain unmodified as there are no major differences. We are grateful to the reviewer for raising these points, and hope that this information helps to strengthen our justifications for the bins in this plot.

19. Line 364: I am not sure I see an order of magnitude change of the velocity. From surface to 1000 m altitude, the velocity changes at most from 0.1 to 0.35 m/s. Please paraphrase for clarity.
Thank you for catching this mistake – it does not vary by an order of magnitude from the time ocean surface to cloud base. We've updated the sentence to say:

"Fall velocities from Figure 4b show that a singular particle's fall velocity can increase significantly from the ocean surface to cloud base."

20. Line 392: Please fix typo to read "meaning they are unlikely to"
Thank you for noticing this mistake, the text has been updated to say the above.

21. Line 439: Please remove "so when" to read "southern shores when SSTs are"
Thank you again for noticing this typo – we've updated the text accordingly.

22. Line 440: Suggest change to read "across a period of one year".
Thank you for this suggestion – we've changed the text to read:

"This study took place on an eastern coastline across a period of one year…."

23. Line 449: The citations should be all in one set of parentheses to read "(Porter and Clarke,1997; Blanchard et al., 1984)"
Thank you for noticing this error – these citations have been put into paratheses.

24. Line 463: Please fix the parentheses to read "Table 2 from Grythe et al. (2014) shows"
Thank you for noticing this error – the citation has been updated to match the above suggestion.

---

## Author Response (AR2)

**Response to Editor**

By Katherine L. Ackerman, Alison D. Nugent, and Chung Taing

Thank you so much for your quick responses and additional care you've taken to review our manuscript and the referees' suggestions. We are incredibly grateful to have someone so aligned with our research to review our paper. In addition to the comments provided by the two referees, we agree with your two suggestions as well.

1) I think Reviewer 1 Comment 4 could maybe be further clarified by reference to Figure 2. As a non-local, I'd also wonder if "Bellows" is the same as "Bellows Beach" cited in the prior study by Porter et al. If the sites are different, please clarify; if the same, it might be useful to explicitly confirm that for those of us not familiar with the sites.

    Yes, Bellows is the same as Bellows beach mentioned in the Porter et al. paper! We've updated this section, as well as linking the map figure to ensure readers understand the locations we're referencing. This section now reads:

    "A linear regression on almost two years' worth of data was conducted between wind speeds measured by a 12-m anemometer at Bellows beach, approximately 7 km northwest of Kaupō Bay (Bellows U12, Fig. 2) and a 10-m anemometer at Mapakpuu beach (WeatherFlow U10, Fig. 2) for trade wind days only (wind directions between 30-90°)."

2) While certainly not required for acceptance, I wonder why the discussion on line 53 of prior work showing a dependence of SSA on SST did not include the recent work showing this for open ocean measurements (Saliba et al. 2019, https://doi.org/10.1073/pnas.1907574116)?

    Thank you for bringing the Saliba et al. paper to our attention – we apologize for overlooking this study previously but recognize it's a substantial piece of work adding to our knowledge of SST and seasonal effects on SSA production. Therefore, we've included this in the SST section of the introduction and are excited to explore the affiliated research in the paper even further!

---

## Author Response (AR3)

**Response to Editor**

By Katherine L. Ackerman, Alison D. Nugent, and Chung Taing

We would like to express our sincere gratitude to our editor of Atmospheric Chemistry and Physics, Dr. Russell, for your remarkable swiftness and unwavering encouragement throughout the publication process of our research article. Additionally, we feel deeply honored to have been chosen as a Highlight Article of Atmospheric Chemistry and Physics. Your prompt responses, insightful feedback, and dedicated support have made this journey both smooth and inspiring. Your commitment to advancing the field and your dedication to nurturing the work of aspiring researchers like myself is truly commendable. We are deeply thankful for your invaluable contributions and look forward to continued collaboration with Atmospheric Chemistry and Physics in the future.